# *Mycobacterium tuberculosis* FasR senses long fatty acyl-CoA through a tunnel and a hydrophobic transmission spine

Julia Lara[1], Lautaro Diacovich [1,2], Felipe Trajtenberg [3], Nicole Larrieux[3], Emilio L. Malchiodi [4], Marisa M. Fernández[4], Gabriela Gago [1], Hugo Gramajo [1✉] & Alejandro Buschiazzo [3,5✉]

*Mycobacterium tuberculosis* is a pathogen with a unique cell envelope including very long fatty acids, implicated in bacterial resistance and host immune modulation. FasR is a TetR-like transcriptional activator that plays a central role in sensing mycobacterial long-chain fatty acids and regulating lipid biosynthesis. Here we disclose crystal structures of *M. tuberculosis* FasR in complex with acyl effector ligands and with DNA, uncovering its molecular sensory and switching mechanisms. A long tunnel traverses the entire effector-binding domain, enabling long fatty acyl effectors to bind. Only when the tunnel is entirely occupied, the protein dimer adopts a rigid configuration with its DNA-binding domains in an open state, leading to DNA dissociation. The protein-folding hydrophobic core connects the two domains, and is completed into a continuous spine when the effector binds. Such a transmission spine is conserved in a large number of TetR-like regulators, offering insight into effector-triggered allosteric functional control.

[1] Laboratory of Physiology and Genetics of Actinomycetes, Instituto de Biología Molecular y Celular de Rosario (IBR-CONICET), Facultad de Ciencias Bioquímicas y Farmacéuticas, Universidad Nacional de Rosario, Rosario 2000, Argentina. [2] Plataforma Argentina de Biología Estructural y Metabolómica (PLABEM), Rosario 2000, Argentina. [3] Laboratory of Molecular and Structural Microbiology, Institut Pasteur de Montevideo, Montevideo 11400, Uruguay. [4] Instituto de Estudios de la Inmunidad Humoral (IDEHU/CONICET-UBA), Facultad de Farmacia y Bioquímica, Universidad de Buenos Aires, Ciudad Autónoma de Buenos Aires 1113, Argentina. [5] Integrative Microbiology of Zoonotic Agents, International Joint Research Unit, Department of Microbiology, Institut Pasteur, Paris 75724 Cedex 15, France. ✉email: gramajo@ibr-conicet.gov.ar; alebus@pasteur.edu.uy

The complex composition of the cell envelope is a distinctive feature of the *Mycobacterium* genus. *Mycobacterium tuberculosis* (*Mtb*) bears peculiar cell wall lipids that play key roles in pathogenicity, also acting as a barrier against environmental stress, antibiotics and the host's immune response[1]. A better understanding of the mycobacterial cell wall biogenesis will likely identify drug targets for the development of new antibiotics, badly needed to combat tuberculosis.

*Mtb*'s outer membrane comprises very long-chain fatty acids (mycolic acids), found in the inner leaflet covalently bonded to the arabinogalactan–peptidoglycan layer, and also in the outer leaflet as non-covalently associated lipids in the form of trehalose mono- and di-mycolates[2]. Mycolic acids, a hallmark of *Mycobacterium*, are synthetised by way of two fatty acid synthase systems, FAS I and FAS II. The multidomain single protein FAS I catalyses de novo biosynthesis of acyl-CoAs in a bimodal fashion rendering $C_{16-18}$ and $C_{24-26}$ derivatives[3]. Long-chain acyl-CoAs are used as primers by the FAS II multiprotein system, and iteratively condensed with malonyl-acyl carrier protein (malonyl-ACP) leading to very long-chain meromycolyl-ACPs (up to $C_{56}$). The latter are eventually condensed to FAS I-synthetised $C_{24-26}$ fatty acids to produce mycolic acids. FAS I-derived long-chain acyl-CoAs are used not only as mycolic acid precursors, but also for the biosynthesis of phospholipids, triacylglycerides, polyketides and other complex lipids, relevant for *Mtb* pathogenicity[4,5]. A complex regulatory network integrating all these pathways must exist in order to maintain lipid homoeostasis. Despite the biological relevance of lipid homoeostasis, little is known about the environmental signals and the regulation cascades controlling lipid metabolism in *Mtb*.

We had previously identified the transcription factor FasR, as a key activator of *fas* and *acpS* gene expression[6]. *fas* and *acpS*, respectively, coding for FAS I synthase and 4-phosphopantetheinyl transferase (essential to produce functional ACP), form a single operon in *Mtb*. FasR activates the transcription of *fas-acpS*, by binding to three inverted repeats in the operon's promoter region[6]. FasR:DNA binding is regulated by long-chain acyl-CoAs, which are themselves products of FAS I, namely acyl-CoAs ≥ $C_{16}$ disrupt the interaction of *Mtb* FasR with its cognate DNA[6]. In addition, FasR is essential for *Mycobacterium smegmatis* viability[6], further highlighting its key role in mycobacterial biology.

FasR sequence reveals homology to members of the TetR family of regulators (TFR), which are one-component sensory transduction proteins[7], typically dimeric and with each protomer displaying a 2-domain all-helical structure[8]. Sequence alignment of FasR with TFRs with known 3D structures (Supplementary Fig. 1a)[9], predicts FasR's architecture to comprise a helix–turn–helix DNA-binding domain towards its N-terminus (residues 1–80), and a larger C-terminal domain corresponding to the ligand- or effector-binding domain (residues 82–228). In contrast to the DNA-binding domain, the effector-binding region reveals little or no sequence homology with other TFRs, such sequence diversity being consistent with the large variety of effector signals sensed by different TFRs[10]. The prototype of the TFR is TetR from the Tn10 transposon of *Escherichia coli*, which regulates the expression of the tetracycline efflux pump in Gram-negative bacteria[11]. However, TFR proteins are widely distributed among bacteria, and control a broad range of processes, including fatty acid biosynthesis[12]. Interestingly, the vast majority of TFRs are transcriptional repressors, with very few exceptions acting as activators[10].

We have now determined the 3D structures of FasR from *Mtb*, in three different states obtained with the protein (i) crystallised alone, (ii) co-crystallised in complex with the fatty acid $C_{20}$-acyl-CoA and (iii) in complex with a double-stranded DNA oligonucleotide bearing the specific FasR-binding sequence. The comparison of these crystal structures, together with the functional characterisation of structure-guided FasR point mutants and molecular dynamics computational simulations, uncovered the molecular mechanisms by which long- ($C_{16}$–$C_{20}$) and very long-chain ($C_{22}$–$C_{26}$) acyl-CoA molecules are sensed by FasR, as well as the means by which such signal disrupts cognate FasR–DNA binding and hence actuates *fas-acpS* transcriptional activation. World-wide efforts have disclosed hundreds of protein structures from *Mtb* corresponding to potential drug targets, a valuable input for a number of drug discovery projects[13,14]. The uncovering of structural and mechanistic insights about a key *Mtb* metabolic regulator, contributes with solid molecular bases for target-based drug discovery, a sensible strategy to combat tuberculosis.

## Results

**Three-dimensional structures of FasR.** Recombinant FasR eluted from size-exclusion chromatography suggesting a dimeric structure (~52 kDa), similar to all TFRs. Attempts to crystallise full-length FasR alone failed. We hypothesised that the 33-amino acid segment at the N-terminus is likely flexible (Supplementary Fig. 1b). Multiple sequence alignments revealed high sequence variation of the N-terminal extensions added to absence of a predicted secondary structure, leading to the construction of a truncated FasR lacking the first 33 amino acids (FasR$_{\Delta 33}$). FasR$_{\Delta 33}$ readily crystallised in the absence of added ligands and also in complex with acyl $C_{20}$-CoA (arachinoyl- or arachidoyl-CoA). Both crystal forms diffracted X-rays at better than 1.7 Å resolution (Supplementary Table 1), and their structures confirm the dimeric architecture of FasR$_{\Delta 33}$, with each protomer organized in two all-helical domains (Fig. 1a), similar to known TFRs[10].

FasR$_{\Delta 33}$-$C_{20}$-CoA was solved using ab initio methods[15]. The structure exhibited the typical TFR architecture, with a DNA-binding HTH (helix–turn–helix) domain from the N-terminus to residue Ser$_{82}$, comprising helices α1–α3. A regulatory effector-binding domain (EBD) is located immediately C-terminal to the HTH, from Lys$_{83}$ to the C-terminus, including helices α4–α9. The latter helices are roughly organized in two bundles, the α4–α7 core runs along the long axis of the ellipsoid regulatory domain, whereas α8–α9, roughly perpendicular to the core, mediate dimerisation by forming a 4-helix bundle with the other protomer's α8′–α9′ helices (Fig. 1a). The FasR$_{\Delta 33}$-$C_{20}$-CoA dimer is strictly symmetric, with the crystallographic twofold axis relating one protomer to the other.

A striking feature of FasR$_{\Delta 33}$-$C_{20}$-CoA is a tunnel-like cavity, delimited by helices α4, α5, α7 and α8, with its two openings towards the bottom and the top of the EBD. $C_{20}$-CoA binds within this tunnel, in a parallel orientation with respect to core helices α4, α5 and α7 (Fig. 1a, b). The tunnel is ~28 Å long, with a predominance of hydrophobic residues on its wall pointing their side chains towards the lumen of the tunnel (Fig. 1b). The fatty acid is well defined all along the tunnel (Supplementary Fig. 2a). Towards the cavity's upper entrance, most of the 4′-phosphopantetheine portion of the CoA cofactor is also observed, the sulfur atom (due to its stronger electron density) was instrumental in positioning the whole $C_{20}$-CoA moiety. Electron density becomes less clear towards the tip of the pantoic group, likely due to high mobility of the CoA portion, eventually vanishing in the region corresponding to the 3′-phosphoadenosine diphosphate group, which were thus not included in the final model.

FasR$_{\Delta 33}$ crystals were also grown in the absence of acyl-CoA with the aim of solving the ligand-free structure. Unexpectedly, additional electron density not corresponding to protein, was visible within the ligand-binding tunnel (Fig. 2a, Supplementary Fig. 2b). It could be part of a polyethylene glycol molecule (PEG 400 was included in the crystallisation mother liquor), but PEG's

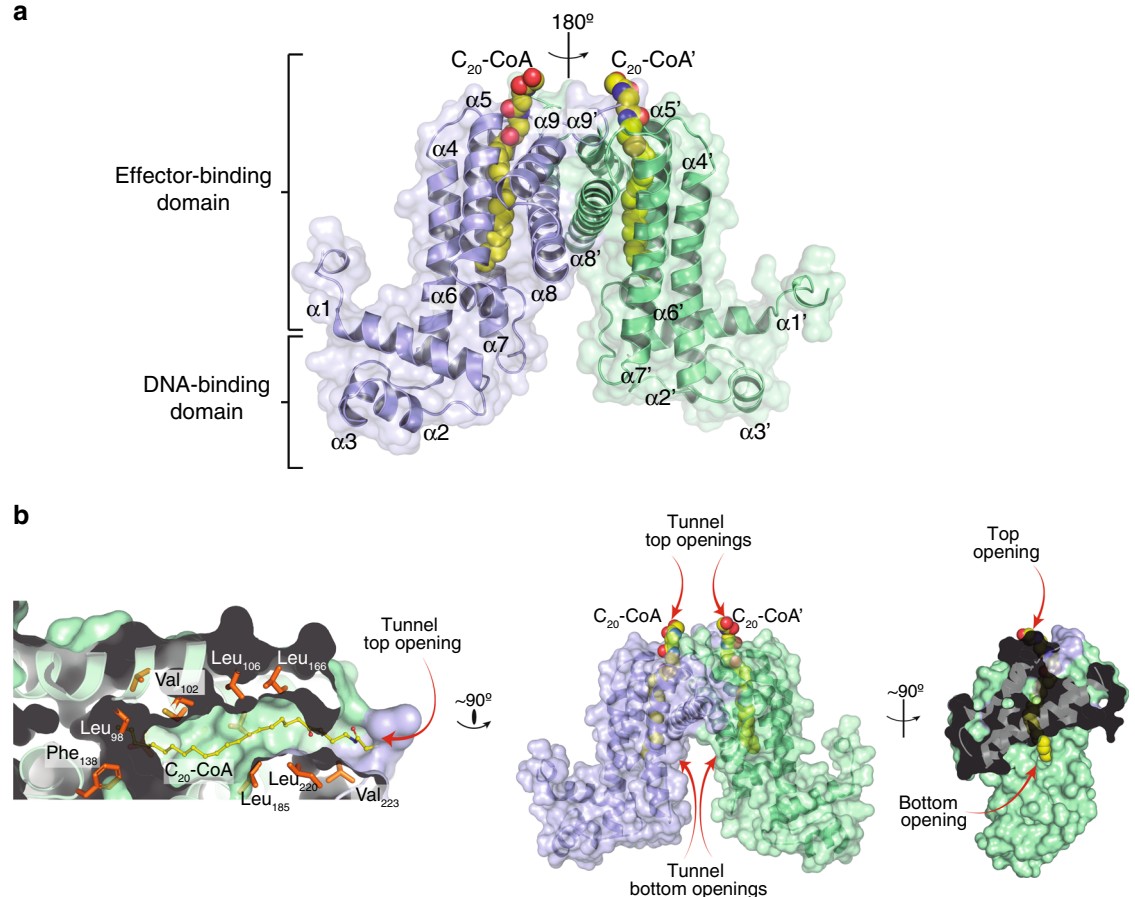

**Fig. 1 Crystal structure of FasR$_{\Delta 33}$-C$_{20}$-CoA. a** FasR dimer in cartoon representation, the two protomers are distinguished in light blue and pale green. The dimer is strictly symmetric, with one protomer in the asymmetric unit, the crystallographic twofold depicted as a vertical axis in the plane of the figure. Secondary structure elements are labelled; a prime symbol denotes equivalent elements in the other protomer. The two domains are indicated, an N-terminal DNA-binding domain with a helix–turn–helix (HTH) motif, and a C-terminal domain with the co-crystallised effector molecule C$_{20}$-CoA bound (in spheres coloured by element). **b** Details of the effector-binding tunnel. Three orthogonal views illustrate the molecular surface of FasR$_{\Delta 33}$.The leftmost cuts along the middle of a protomer, revealing the top tunnel mouth on the right and a large segment of the tunnel itself with the bound acyl moiety (in sticks). A few amino acid side chains that define the tunnel walls are labelled. The rightmost panel is an open-book perspective, with the footprint of protomer A visible on protomer B's surface (in pale green), the two openings of the tunnel are visible, showing solvent exposure of both tips of the C$_{20}$-CoA molecule.

bridging oxygens are energetically costly if buried within the tunnel's hydrophobic environment. The density is also consistent with myristic acid (C$_{14}$), which we hypothesize could likely bind during protein expression in *E. coli*. Even if the chemical nature of the bound species is not certain, this piece of evidence results in two consequences: FasR$_{\Delta 33}$-C$_{14}$ is not a true apo form of the protein, but it did disclose by serendipity the structure of FasR with a shorter alkyl chain bound in the effector pocket, as compared with FasR$_{\Delta 33}$-C$_{20}$-CoA.

In contrast to the strictly symmetric organization of the FasR$_{\Delta 33}$-C$_{20}$-CoA dimer, FasR$_{\Delta 33}$-C$_{14}$ displays one full dimer per asymmetric unit (Fig. 2b), each protomer deviates from strict symmetry with respect to the other. A strong twofold non-crystallographic operator relates nonetheless both protomers, but applying only to the regulatory EBDs. The HTH domains of FasR$_{\Delta 33}$-C$_{14}$ depart from this relationship, after superimposing one protomer onto the other (Fig. 2b), the EBDs fit together well, while the HTHs are rotated by ~15°. To further characterise this symmetry deviation, two identical FasR$_{\Delta 33}$-C$_{14}$ dimers were superimposed by maximising the fit between one HTH domain from distinct protomers on each dimer (0.3 Å root-mean-square deviation (rmsd) considering all atoms of the two superimposed HTHs). This rotation operation resulted in >5.5 Å rmsd between

the other pair of HTHs. A similar exercise using the EBDs revealed a considerably smaller difference, 0.6 Å on the super-imposed pair vs 1.5 Å for the other. Departure from intradimer symmetry is thus largely due to substantial flexibility in the region that joins the regulatory and the DNA-binding domains, not the hinge loop covalently linking both domains, but rather the whole region involving the HTH as a rigid domain plus the lower part of the regulatory domain's helices that interact with the HTH (mainly the N-terminal half of α4). Such type of flexibility mimics the swinging of a pendulum (Fig. 2b, c), and is consistent with higher atomic displacement parameters and weak electron density in the α6–α7 loop as well as in the N-terminal half of helix α7, features that were not apparent in FasR$_{\Delta 33}$-C$_{20}$-CoA (which displayed more rigidity including in the HTH domains).

The dimerisation interface is an extremely well conserved structural feature among the entire TFR[10], always involving a helical bundle constituted by equivalent helices, α8 and α9 from each protomer (according to FasR helix numbering scheme). FasR$_{\Delta 33}$-C$_{14}$ and FasR$_{\Delta 33}$-C$_{20}$-CoA dimers were superimposed maximising the fit between the corresponding dimerisation helix bundles. Clear differences between both structures were revealed (Fig. 2c). In FasR$_{\Delta 33}$-C$_{20}$-CoA, the dimer is opened up, with both protomers separating away from each other, compared with

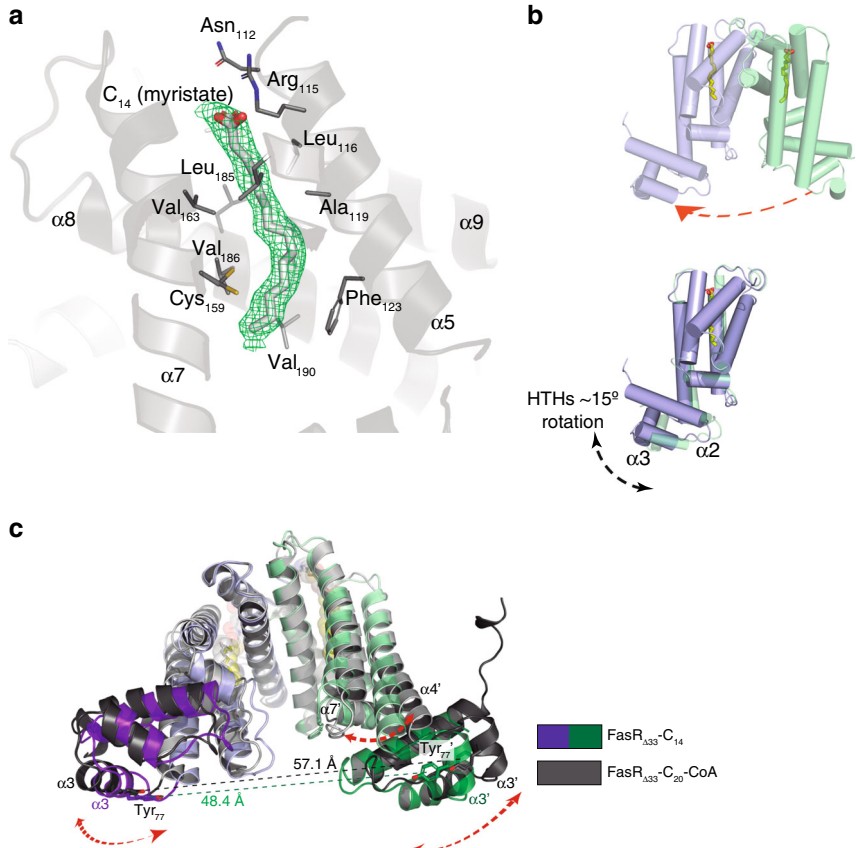

**Fig. 2 FasR$_{\Delta 33}$-C$_{14}$ exhibits larger protein flexibility including dimer asymmetry. a** SigmaA-weighted mF$_{obs}$-DF$_{calc}$ difference Fourier map contoured at 3.5$\sigma$ (green mesh) calculated with no ligand bound in the effector-binding tunnel during FasR$_{\Delta 33}$-C$_{14}$ refinement. The FasR$_{\Delta 33}$-C$_{14}$ model is shown with grey cartoons, and residues at $\leq$4 Å from the ligand are depicted with thin sticks. The electron density allowed to model a myristic acid, overlaid within the difference map in thick sticks coloured by atom. Note polar and charged residues on the top opening, close to the carboxyl head of the ligand. Hydrophobic residues line up the rest of the tunnel's walls. **b** FasR$_{\Delta 33}$-C$_{14}$ protomers A and B are coloured as in Fig. 1. The dimer is asymmetric, quantitated by superimposing protomer B onto A (red arrow in the top panel), after what the effector-binding domains fit quite well while the HTH domains show significant rotation between one another (bottom panel), with max root-mean-square deviation (rmsd) on helices α2 and α3. **c** The FasR$_{\Delta 33}$-C$_{20}$-CoA and FasR$_{\Delta 33}$-C$_{14}$ structures are superposed, resulting in maximal fit on the central 4-helix bundle that mediates dimerisation (helices α8–α9 from both protomers). The upper portion of the effector-binding domains end up well aligned, but significant shifts affect the lower part together and the HTH domains (dashed arrows). The distance between the centres of mass of Tyr$_{77}$ on helix α3, differs by more than 10 Å between both structures. The HTH domains move consolidated with the bottom portion of juxtaposed helices α4 and α7 from the effector-binding domains.

FasR$_{\Delta 33}$-C$_{14}$. Using the position of conserved Tyr$_{77}$ Cα as a reference (at the centre of helix α3, key to mediate DNA-binding), the two HTH domains in FasR$_{\Delta 33}$-C$_{14}$ are ~48 Å apart, vs ~57 Å in FasR$_{\Delta 33}$-C$_{20}$-CoA (Fig. 2c). Taking the above observations together, we hypothesize that binding of a ligand, long enough to fill up the entire hydrophobic tunnel, stabilises the FasR dimer in a more rigid configuration, fixing an open HTH geometry that is incompatible to bind DNA. In contrast, when the tunnel is occupied with a too short acyl chain, greater HTH mobility is enabled, allowing helices α3 to reach a proper separation distance to binding DNA (ideal B-DNA has 34 Å separation between major grooves on the same side of the double helix). This flexibility cannot be described as a simple rigid-body movement of one domain with respect to the other (a hinge-like interdomain flexibility), but rather as a rearrangement implicating the HTH and part of the EBDs acting as a consolidated assembly.

**Acyl-CoA-binding effect on FasR–DNA association**. To test the structure-based hypotheses about acyl binding and its effect on FasR–DNA association, point mutants were designed to block the entrance of the ligand into the hydrophobic tunnel. Tunnel-blocking was expected to abolish ligand-triggered rigidification of

the protein and consequent DNA-binding hindrance. Two mutants were constructed: FasR$_{L106F}$ substitutes Leu$_{106}$ by a phenylalanine at the entrance of the tunnel; whereas FasR$_{LVL}$ adds bulky side chains not only on position 106, but also substituting Leu$_{185}$ and Val$_{163}$ by phenylalanines (Fig. 3a). Point mutations did not affect the dimeric architecture of FasR$_{L106F}$ (Supplementary Fig. 3), and while slightly <50% of the triple mutant FasR$_{LVL}$ eluted as a monomer, >50% behaved as the wild-type protein.

Electrophoretic mobility shift assays (EMSAs) were performed by pre-incubating FasR, FasR$_{L106F}$ and FasR$_{LVL}$ (its dimeric form) with C$_{16}$-CoA and C$_{20}$-CoA, and then incubating these reactions with $^{32}$P-labelled *fas* promoter (P*fas*$_{MT}$). Strongly supporting our hypothesis, the acyl-CoA ligands triggered bare or not detectable DNA dissociation in the case of FasR$_{L106F}$ and FasR$_{LVL}$ mutants, while clearly inhibiting DNA-binding of wild-type FasR (Fig. 3b). FasR–DNA apparent dissociation constants ($^{app}K_D$) were not significantly affected by the point mutations (Supplementary Fig. 4a). Although direct monitoring of acyl-CoA binding to FasR$_{L106F}$ and FasR$_{LVL}$ mutants was not feasible due to technical impediments, dose–response analyses by EMSA (Supplementary Fig. 4b) further confirmed that the ligands likely do not bind to the

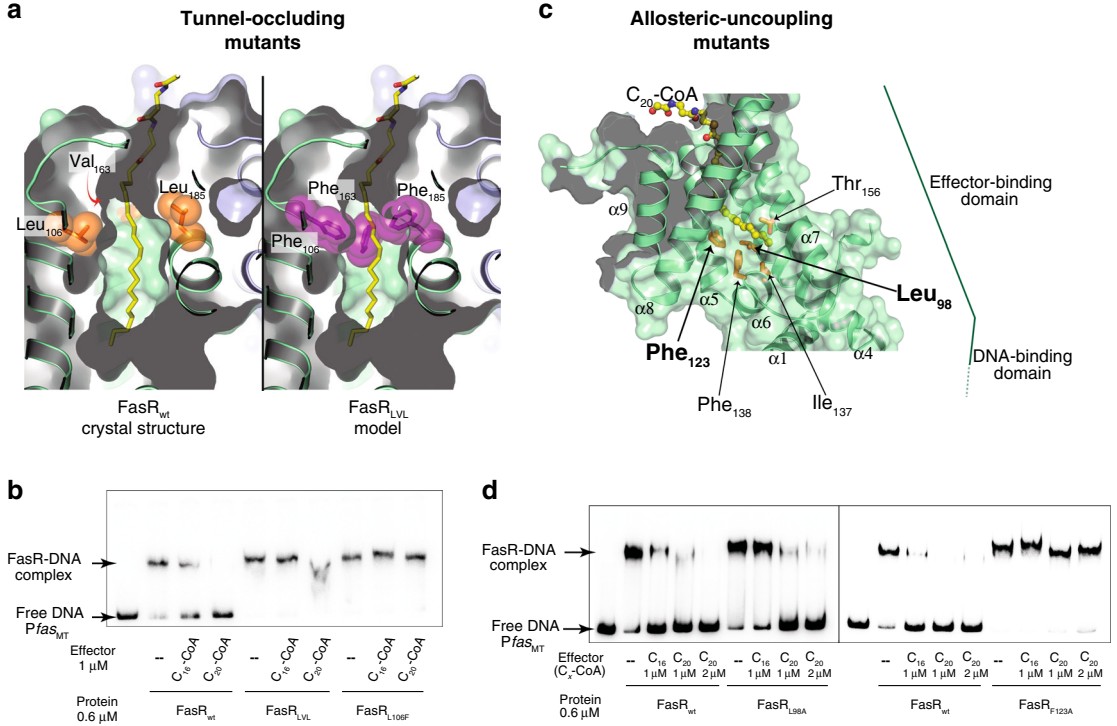

**Fig. 3 Structure-guided mutations in FasR leading to permanent DNA-binding. a** Selected mutations to occlude the effector-binding tunnel are illustrated. The crystal structure of FasR$_{\Delta33}$-C$_{20}$-CoA (left panel) is compared to a model of the triple mutant FasR$_{LVL}$ (right panel). Val$_{163}$ is actually behind the tunnel (red arrow). The substitutions by bulky phenylalanines anticipate steric hindrance of the acyl chain (shown as sticks coloured by element). **b** Electrophoretic mobility shift assay was performed by incubating the $^{32}$P-labelled 398 bp P$fas$ promoter region with wild-type FasR and mutant FasR proteins (FasR$_{LVL}$ and FasR$_{L106F}$) in the absence or in the presence of C$_{20}$-CoA and C$_{16}$-CoA. Protein–DNA complexes were separated by electrophoresis on a 6% polyacrylamide gel. The assay was performed in duplicate, producing similar results. Source data are provided as a Source Data file. **c** Selected mutations to uncouple allosteric signalling from the ligand-binding pocket to the DNA-binding domain. The bottom mouth of the effector-binding tunnel is located towards the centre of the molecule, revealing the tip of C$_{20}$ (in ball and stick representation, coloured by atom). A few side chains are labelled and shown as orange sticks, chosen among residues bordering the tunnel opening. Two bulky residues (Leu$_{98}$ on helix α4 and Phe$_{123}$ on α5) that were substituted by alanines, are labelled with bold fonts. **d** Gel shift assays of the wild-type and selected FasR mutants. The DNA probe that corresponds to the 398 bp P$fas$ promoter region (P$fas_{MT}$) was $^{32}$P-labelled and incubated either with FasR wild-type, FasR$_{L98A}$ or FasR$_{F123A}$, in the absence or in the presence of the indicated concentrations of C$_{16}$-CoA and C$_{20}$-CoA. The assays were performed in duplicate, producing similar results. Source Data are provided as a Source data file.

tunnel-occluding mutants. While FasR$_{wt}$ displayed nanomolar-range half-maximal inhibitory concentrations (IC$_{50}$ = 368 nM) and apparent inhibition constants ($^{app}K_i$ = 42 nM) with C$_{20}$-CoA, the tunnel-blocking mutants exhibited significantly increased values. Namely, FasR$_{L106F}$ showed dissociation of the protein–DNA complex only at the highest acyl-CoA concentrations (>4 μM), and FasR$_{LVL}$ remained bound to DNA even with 6 μM C$_{20}$-CoA (Supplementary Fig. 4b). Ligand entrance within the tunnel is thus needed in order to trigger the protein conformational change that precludes binding of the regulator to its cognate DNA site.

**Allosteric mechanism of signal transmission.** The shift of the bottom-half of helices α4 and α7, comparing FasR$_{\Delta33}$-C$_{14}$ and FasR$_{\Delta33}$-C$_{20}$-CoA structures (Fig. 2c), strongly suggested that long enough alkyl chains in the effector-binding tunnel are critical in triggering the rigidification rearrangement, which results in arm-opening. To understand the molecular bases for such effect, the protein region around the distal tip of the ligand acyl chains were analyzed in detail. The very last carbon atoms of C$_{20}$-CoA interact with mostly bulky hydrophobic residues towards the end of the tunnel, i.e. at the bottom opening of the tunnel that leads to the space separating both protomers in the dimer. Among these hydrophobic residues Leu$_{98}$ (on helix α4), Phe$_{123}$ (on α5) and Phe$_{138}$ (on α6) might play relevant roles (Fig. 3c). The substitution

of such voluminous hydrophobic residues by smaller alanine side chains, could uncouple the HTH mobility-restraining effect from ligand-binding. Point mutants FasR$_{L98A}$ and FasR$_{F123A}$ were constructed, which maintained a normal dimeric structure (Supplementary Fig. 2). Both mutants showed significant functional effects, uncoupling ligand-binding and DNA association (Fig. 3d), with a clearer effect observed in the case of FasR$_{F123A}$. To further dissect the underlying mechanisms, DNA-binding affinities were analyzed, comparing wild-type vs FasR$_{L98A}$ and FasR$_{F123A}$ mutants (Supplementary Fig. 5). Dissociation constants were quantitated from EMSA data, all in the nanomolar range (Table 1). Compared with FasR$_{wt}$, FasR$_{L98A}$ displayed slightly lower affinity for the P$fas_{MT}$ probe and FasR$_{F123A}$ higher, but neither one exhibiting significant effects. According to the hypothesis that these point mutations would, however, affect efficient transmission of the signal from the effector-binding to the DNA-binding domain, we next calculated IC$_{50}$ and $^{app}K_i$ values for the three proteins by evaluating dose–response effects with increasing concentrations of C$_{20}$-CoA (Supplementary Fig. 6). Indeed, both mutants had significantly lower FasR–DNA-dissociation responses to the ligand (Table 1), particularly so for FasR$_{F123A}$. The length of the acyl chain was also critical, with IC$_{50}$s and $^{app}K_i$s all shifted to significantly higher values when C$_{16}$-CoA was used (Supplementary Fig. 7 and Table 1). These results strongly suggest that residues Leu$_{98}$ and Phe$_{123}$, especially the latter, are key to ensure allosteric signal

**Table 1 FasR association to DNA, and dose-dependent inhibition by effector ligands.**

|  | $FasR_{wt}$ | $FasR_{L98A}$ | $FasR_{F123A}$ |
|---|---|---|---|
| **Binding to DNA ($Pfas_{MT}$)** | | | |
| $^{app}K_D$ (nM) | $83 \pm 11$ | $155 \pm 18$ | $39 \pm 6$ |
| **$C_{20}$-CoA dose response** | | | |
| $IC_{50}$ (nM) | $368 \pm 105$ | $634 \pm 100$ | $5269 \pm 1067$ |
| $^{app}K_i$ (nM) | 42 | 152 | 571 |
| **$C_{16}$-CoA dose response** | | | |
| $IC_{50}$ (nM) | $5415 \pm 1047$ | ND | ND |
| $^{app}K_i$ (nM) | 589 | ND | ND |

Comparison of $FasR_{wt}$ vs mutants $FasR_{L98A}$ and $FasR_{F123A}$ that uncouple the allosteric effect. $^{app}K_D$ and $IC_{50}$ values are reported as the average of at least three independent experiments ± one standard error of the mean.
ND: not detectable.

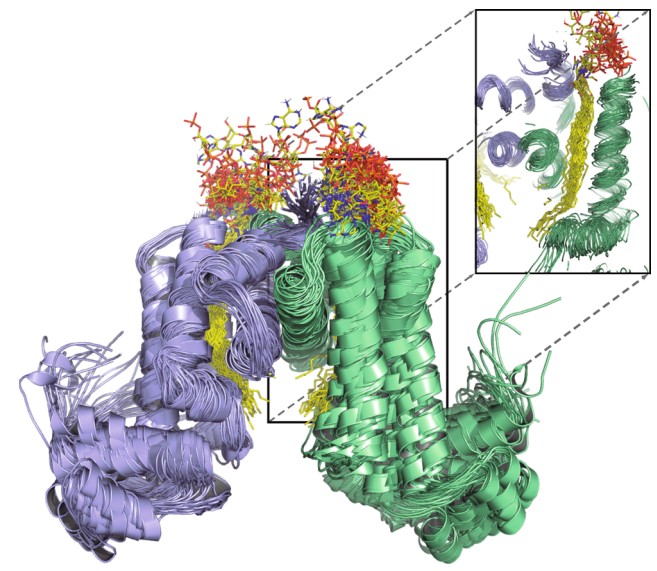

**Fig. 4 FasR can accommodate longer than $C_{20}$ acyl chains.** A representative 10 ns trajectory of all-atom molecular dynamics is illustrated by showing in cartoon representation the simulated model of $FasR_{\Delta 33}$ in complex with $C_{26}$-CoA every 250 ps (see Supplementary Movie 1). Shown as superposed ensemble, the secondary structure elements of both domains are preserved, and the cerotic acyl chain (in sticks) also remains stable within the tunnel. The inset shows a projection of a cut section through protomer's B tunnel, to display the dynamic stability of the dimeric interface and the acyl chain (as opposed to the large flexibility of the nucleotide portion of coenzyme-A). Free volume is available at the interprotomer space, predicting that even longer acyl moieties should be able to accommodate. See Supplementary Fig. 9 for rmsd values within and between domains, comparing also the simulated behaviour of apo-FasR.

transmission while not influencing DNA-binding. That these mutations uncouple signal transmission but do not alter the protein's affinity for the effector ligand was assessed for the $FasR_{F123A}$ mutant (Supplementary Fig. 8). Surface plasmon resonance showed comparable association kinetics of $FasR_{F123A}$ to $C_{20}$-CoA as compared to $FasR_{wt}$. In sum, specific residues that are not essential to bind acyl ligands nor DNA, play a key role in transmitting the signal between both domains of the protein once the effector-binding tunnel is fully occupied.

**Very long fatty acyl binding anticipates FasR rigidification.** Very long fatty acyl moieties are relevant in the biology of Mycobacteriaceae including *Mtb*[16]. Intermediates in the synthesis, and constitutive moieties, of mycolic acids, very long fatty acids are essential components of mycobacterial cell walls. How can such long alkyl chains act as effectors of FasR? A fully extended $C_{20}$ acyl chain measures 26.5 Å, within the FasR-$C_{20}$-CoA tunnel the acyl shows some bending reducing that length to ~20 Å. The latter magnitude is enough for the $C_{20}$ chain to fully occupy the tunnel, its distal tip located immediately beside the bottom opening that leads to the space between protomers in the dimer. We predict that longer acyl chains will be able to accommodate, protruding additional carbon atoms into the interprotomer space. To provide support for this scenario, classical molecular dynamics trajectories were calculated (Supplementary Movie 1) starting from our FasR-$C_{20}$-CoA structure where the $C_{20}$-CoA was substituted in silico by the ~8 Å longer $C_{26}$-CoA (cerotic acyl-CoA), a particularly important biosynthesis intermediate synthesised by FAS I to provide for the α-alkyl chain of mycolic acids. After initial energy minimisation, 10 ns all-atom trajectories were simulated with explicit solvent, showing that the cerotic acyl chains are stable within the tunnel, their 6-carbon extensions indeed protruding into FasR's interprotomer space (Fig. 4). The available volume in the open form of acyl-bound FasR anticipates even longer acyl chains to be readily accommodated, considering that the protein shows a very stable behaviour with ≤2 Å rmsd within its effector-binding and DNA-binding domains (Supplementary Fig. 9a). Similar trajectories were simulated in the absence of bound effector, suggesting that the effector-binding domain becomes less stable, also exhibiting a larger wiggling of the HTH domains as revealed by calculating their rmsd with the EBDs superimposed (Supplementary Fig. 9b).

**The FasR–DNA 3D structure confirms the allosteric mechanism.** The crystal structure of full-length FasR was solved by co-crystallisation with a 25-bp double-stranded oligonucleotide bearing the native FasR-binding sequence motif (Fig. 5). A number of crystals and cryo-protections methods were tested, consistently producing strongly anisotropic X-ray diffraction data, reaching 3.85 Å resolution in the best direction (Supplementary Table 1). A form of crystal disorder affected the position of the DNA double helix (details in Methods), occupying equivalent positions in different unit cells while sitting alternatively in both 5′→3′ directions according to a crystallographic twofold.

The protein that was used to grow the FasR–DNA crystals corresponds to full-length FasR. However, the first ~30–35 amino acids at the N-termini are not visible in electron density, indicating they are not bound to the DNA, likely due to high flexibility. Neighbouring FasR dimers related by crystallographic symmetry, are observed bound to the same DNA fragment on a juxtaposed consecutive manner, at roughly 90° one from the other. Such organization is associated to a pronounced bending of the DNA molecule. The position of the major DNA kink is similar to the one identified in other TFRs, inducing a very similar bending angle as in TetR[17], or yet similar in magnitude but inverted compared to the archaeal FadR[18] due to the significantly shifted positions of the HTH domains relative to the dimeric effector-binding core (Supplementary Fig. 10). The limited resolution of the FasR–DNA structure, and the presence of crystal disorder affecting the occupancy of the DNA molecules, precluded detailed analyses of protein:DNA interactions. However, the structure did provide two accurate pieces of evidence: (i) the absence of ligand bound within the tunnel of the effector-binding domain, and (ii) a major conformational rearrangement bringing the HTH DNA-binding domains closer together in the dimer (Supplementary Movie 2) such that helices α3 now fit within two successive major grooves on the DNA molecule.

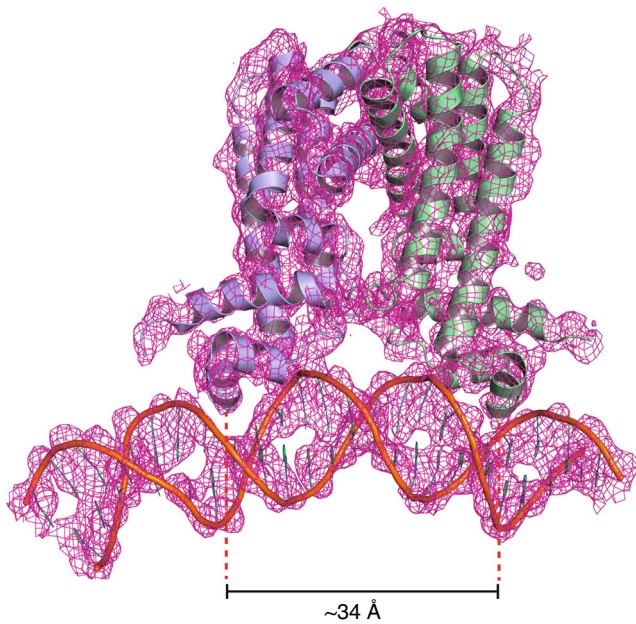

~34 Å

**Fig. 5 Crystal structure of full-length FasR in complex with DNA.** Electron density map (sigmaA-weighted $2mF_{obs}-DF_{calc}$ Fourier) is shown (magenta mesh), overlaid on the final refined model, showing the protein dimer (protomers in light blue and pale green) and the DNA double helix (orange) in cartoon representation. The map was carved around the atomic model with a border of 3 Å to improve the clarity. Helices α3 are seen fitting within the DNA major groove as expected.

This conformational change hampers effector occupation within the tunnel, the latter seems to be constricted by shifted residues (e.g. $Phe_{123}$) which move their side chains towards the tunnel's lumen. The $FasR_{\Delta33}$-$C_{14}$ structure described above proved that acyl-containing compounds from *E. coli* are invariably associated within the effector-binding cavity, even if specific acyl/acyl-CoA molecules are not added during protein purification and crystallisation. That the FasR tunnel in the FasR–DNA complex is free of bound ligands is supported by unequivocal evidence from difference Fourier maps at both early and late stages of refinement (detailed in Methods). The association to DNA thus correlates to expelling ligands from the effector-binding tunnel of FasR, at least those that attach more loosely within the cavity.

## Discussion

FasR is a TFR member that senses long and very long acyl-CoA moieties, subsequently turning off FAS I-mediated fatty acid biosynthesis. *Mtb* is able to synthesise very long acyl-CoA intermediates (e.g. in the way to synthesising mycolic acids), spanning molecular lengths of 40–50 Å and more. It is known that TFRs possess pockets, sometimes even deeper tunnel-like cavities, which bind to, and enclose the sensed effector ligand. How can FasR deal with the very long effector molecules it senses? It must be stressed that the entire effector-binding domain of FasR measures ~40 Å along its longest axis (and substantially less considering the inner cavity): hence, the fatty acyl effectors can often be longer than the protein's own physical boundaries. We now answer to this question by revealing a unique hydrophobic tunnel that cuts across the entire effector-binding domain of FasR (Fig. 1), a tunnel that is conspicuously opened on both ends. This singular solution has evolved to lodge the kind of acylated chains of 20 and more carbons that inhibit FasR binding to its cognate DNA[6].

In addition to the tunnel itself, and in continuity with the bottom opening of it, FasR possesses a cavity delimited by the two protomers. The volume of such cavity[19] in FasR is unusually large (~1800 Å³) compared to many other TFRs: ~600 Å³ (*S. enterica* RamR PDB 3VVY and *M. tuberculosis* EthR PDB 5NIO), ~530 Å³ (*E. coli* RutR PDB 4XK4), ~340 Å³ (*E. coli* TetR PDB 2XPW), ~135 Å³ (*S. acidocaldarius* FadR PDB 6EL2), or yet 40 Å³ (*P. aeruginosa* DesT PDB 3LSJ). This feature is likely relevant, as it anticipates FasR's ability to accommodate very long fatty acyl chains, such as $C_{26}$-CoA (synthesised by the *Mtb* FAS I system[20]), while maintaining a stable, open configuration (Fig. 4, Supplementary Movie 1). Other fatty acid-sensory TFRs fairly similar to FasR[18,21–23], either do not create true continuous tunnels[22], or engage a different set of residues running in a perpendicular direction as compared with FasR's cavity[18,21,23]. EthR is yet another TFR from *Mtb*[24], intensively investigated as a target to develop anti-tuberculosis medicines. However, EthR is substantially different from FasR (22% sequence identity; ~4.5 Å rmsd after superposition of the effector-binding domains), with a shorter N-terminal extension before the first α helix, and a tunnel displaying wider regions or bulges (Supplementary Fig. 11). Such bulges seem to correlate with EthR's capacity to bind compounds that include 1–3 aromatic or aliphatic rings[25]. At difference with FasR, the physiologic molecules sensed by EthR remain unidentified, despite the >70 EthR crystal structures available, most in complex with surrogate ligands. Among these, only one corresponds to a linear chain (hexadecyl octanoate, in PDBs 1U9N and 1U9O), in this way the most similar to FasR effectors. This ligand is positioned in a similar configuration as the acyl ligands in FasR, but leaving part of EthR's available tunnel volume unoccupied (Supplementary Fig. 11c), suggesting that physiologic effectors are likely to be of larger size, branched and/or containing bulkier cyclic groups.

FasR is thus equipped to binding very long-chain acyl effectors, but how is such binding coupled to inhibiting DNA association? The type of rearrangements that we have found (Supplementary Movie 2) are consistent with the ones observed in a number of other TFRs, in principle conforming to the mechanistic hypothesis that the binding of effector ligand (the signal) induces an HTH-open conformation, eventually inhibiting TFR association to DNA (the output response). Association of TFRs to DNA indeed requires the HTH domains to close in, in order for the α3 helices of the two protomers to fit into two successive major grooves (~34 Å apart) on the same side of the cognate DNA[26]. Such a mechanism has been put forward to explain the workings of *E. coli* TetR when binding tetracycline[27], DesT from *Pseudomonas aeruginosa* sensing saturated vs unsaturated acyl-CoAs[22], RutR recognizing uracil in *E. coli*[28] or yet the multi-drug binding protein QacR from *Staphylococcus aureus*[29], among many others.

Binding the effector stabilises an open configuration of FasR, which does not necessarily imply that the effector mechanically triggers a closed to open transition. If the latter mechanism were true, the ligand-free structure should exhibit a closed, DNA-binding competent configuration. A ligand-free form of FasR could not be crystallised, but turning our attention to the vast number of available TFR crystal structures, those with no effector bound, often correspond to the open form[8,10], contradicting the predicted outcome. Among the few apo TFRs that exhibit closed configurations, several reveal crystal packings that fortuitously fix the HTH domains strongly in place (e.g. PDB IDs 2FX0, 1T33, 3VOX and 4JKZ, among others). Moreover, by comparing some of these apo crystals with their DNA-bound counterparts (e.g. 4JKZ vs 4JL3), substantial shifts of the HTH domains can readily be observed, revealing a closed-like configuration, but shifted with respect to the proper DNA-binding-competent one. Reliable information about the true closed configuration has thus largely

been obtained from crystal structures of TFRs in complex with DNA. Additional evidence further challenge a simple open/closure mechanism: (i) no obvious positional shifts of individual residues can explain the mechanical bases of the alleged pendular movement; (ii) a number of TFR mutants have been identified that either uncouple effector-binding from transcriptional induction[30], or invert the effector's action by triggering a tighter binding to DNA[31,32], in both cases often implicating amino acid residues not directly involved in effector-binding.

Finite deformations physics theory seems attractive to highlight allosteric regulation pathways by measuring mechanical strain rather than pairwise atomic position deviations[33]. Unexpectedly, one of the segments subjected to highest mechanical strain in all TFRs analyzed, corresponds to the loop that connects helices α6 and α7 (Supplementary Fig. 12). A triangle defined by α5, α6 and α7, a conserved feature in all TFRs[8], harbours the ligand-binding core cavity (which can expand into tunnels with top, bottom and/ or lateral openings). Helix α6, associating to α8, is attached to the fixed core, upper-half of the effector-binding domain; but simultaneously, α6's C-terminal tip and the α6–α7 junction also associate to the moving HTH. In sum, the α6–α7 loop is bound to fixed and moving parts, eventually leading to local deformation. Residues that might explain this strain, in contact with helices α6, the α6–α7 loop and the HTH domain, compose an array of hydrophobic residues that is highly conserved among TFRs (Supplementary Fig. 1 and Supplementary Data 1 and 2), configured in three-dimensions as a continuous spine connecting the two domains of FasR. This spine belongs to, and connects the hydrophobic protein-folding cores of both domains, being interrupted by the ligand-binding cavity in all TFRs analyzed (Fig. 6 and Supplementary Fig. 13). Only in the ligand-bound condition this hydrophobic spine is completed, by the ligand molecule itself at the effector-binding domain, stabilising a rigid and open conformation. Such a mechanism predicts a disordered (flexible) to ordered transition of the TFR protein, which is consistent with the evidence we provide for FasR as well as with available evidence from other TFRs[8,10,30–32,34]. In particular, fluorescent probes that bind to partially folded proteins in molten globule states[35], have been shown to bind promiscuously to apo TFRs[8]. Also, effector-triggered appearance of folding cooperativity between both domains, as well as proteolysis-resistance, have been reported in wild-type and not in allosteric-uncoupled mutants of TetR[34,36]. Indeed, full-length FasR became more resistant to trypsin proteolysis when pre-incubated with $C_{20}$-CoA (Supplementary Fig. 14), consistent with achieving a more compact fold. Taken together, the extensive body of evidence lends strong support to the transmission spine mechanism as the most consistent interpretation of the effector-mediated allosteric control of TFRs' DNA-binding function. We cannot, however, exclude that several distinct regulatory mechanisms might have evolved in different subsets of the superfamily, correlated with the broad range of sequence variation of effector-binding domains.

The transmission spine mechanism implies that when the ligand leaves the site, or if it is too short to fully occupy it, the hydrophobic spine is broken, protein folding is sub-optimal and a multitude of conformations of the HTH domain is anticipated (HTH wiggling is illustrated schematically in Fig. 7), including conformations that are competent for DNA-binding. In this line of reasoning, a $C_{14}$ acyl moiety does not trigger a full-blown disorder-to-order transition, revealed by dimer asymmetry and higher protein flexibility as observed in the FasR$_{\Delta 33}$-$C_{14}$ complex. This hypothesis also explains why mutating bulky residues that contribute to building and stabilising the spine (e.g. Leu$_{98}$ and Phe$_{123}$ in FasR), can uncouple the allosteric effect: the ligand is then insufficient to achieve a complete, compact fold in the mutated form (Supplementary Fig. 15). Residues that are not directly involved in effector-binding, but that contribute to building and/or stabilising the spine, will also be able to exert notable effects on allosteric coupling, upholding reported results[30–32].

A second conformation, the one bound to DNA, is, however, compactly folded. In this case it is the polynucleotide that pulls on the flexible HTH domains of the dimer bringing them closer together. Correlated to this HTH movement and again transmitted through the hydrophobic spine, several of the bulky residues that line up the effector-binding cavity walls, occlude the cavity (such as Phe$_{123}$ in FasR–DNA), completing the spine into a "folded protein-like" core. This occlusion of the ligand-binding pocket in other TFRs when bound to DNA has been described in crystal structures of TetR[29], DesT[22] and FadR[23].

This scenario opens up exciting avenues to be explored. How reversible is the binding of effector compounds, and what triggers their dislodging from the cavity? DNA-binding might likely expel the effector and vice-versa, depending on relative DNA-vs-effector concentrations and affinities. Of note, the assistance of other proteins, in burying the long alkyl moieties when expelled from FasR and other lipid-sensing TFRs should not be ruled out, as they might be significant components of the thermodynamic equations. The hydrophobic transmission spine hypothesis (Fig. 7) may prove instrumental to design better drugs. Eukaryotic protein kinases (ePKs) are not homologous to TFRs, but do exhibit two analogous hydrophobic spines connecting the two domains of ePKs, regulating their activation switch and catalysis[37]. The spine-mediated regulatory mechanism has been successfully exploited to develop ePK-targeted drugs against cancer and inflammatory diseases[38,39]. A closer example concerns anti-tuberculosis drug discovery. Comparing EthR structures co-crystallised with different inhibitors[40]. Compounds simultaneously bearing thienyl and piperidinyl pharmacophores were selected as the most potent among the screening hits. The transmission spine offers a mechanistic explanation as to why the piperidinyl-binding pocket is the crucial region to improving inhibitory activities[40] in the hit-to-lead development. The piperidinyl- and not the thienyl-interacting pocket engages EthR hydrophobic-spine residues (PDBs 3G1O, 3G1M). Structure-guided drug discovery strategies that exploit the allosteric hydrophobic-spine transmission mechanism might thus prove successful in developing novel medicines against tuberculosis, including multi- and extensively drug-resistant strains.

## Methods

**Bacterial strains and plasmids.** A summarised list of primers, plasmids and cell strains is included as Supplementary Notes in the Supplementary Information. *E. coli* strain DH5α cells were used for DNA cloning purposes, transformed according to standard methods. The *E. coli* BL21 CodonPlus(DE3)-RIL and BL21 λ (DE3) strains were used instead for protein expression. The *fasR* gene (*rv3208*) was PCR-amplified from *Mtb* H37Rv genomic DNA using the oligonucleotides F-TevRv3208 (5′-CCCTCCATATGGGAAAACCTGTACTTCCAGGGTATGAGCGATCTCGCC AAG-3′) to introduce an NdeI site at the translational start codon and encode a fused Tobacco Etch Virus protease digestion site (TEV), and R-Rv3208 (5′-GA ATTCCTACGAGCGGGTAAGCG-3′) to introduce an EcoRI site at the end of the ORF. To generate a FasR recombinant protein with a hexa-histidine-TEV-tagged site at the N-terminus, the corresponding PCR product was extracted from the agarose gel, cloned into the pCR BluntII TOPO vector (Invitrogen), digested with NdeI and EcoRI, and finally subcloned into the expression vector pET28a as described by the manufacturer (Merck). The recombinant plasmid (pET28_ *fasR*) was transformed into BL21 CodonPlus(DE3)-RIL cells, and transformants were selected on LB agar plates containing 50 µg/ml kanamycin. To produce the His-tagged FasR version used in EMSA and controlled proteolysis experiments (FasRwt), *fasR* was PCR-amplified from genomic DNA of *Mtb* H37Rv using the oligonucleotides F-Rv3208 (5′-CATATGAGCGATCTCGCCAAGACA-3′) to introduce a NdeI site at the translational start codon of *fasR* gene, and R-Rv3208 (5′-GAATTCCTACGAGCGGGTAAGCGG-3′) to introduce an EcoRI site at the end of the ORF. To generate a *fasR* His-tag fusion gene, the PCR product was cloned into the pCR BluntII TOPO vector, digested with NdeI and EcoRI and subcloned into NdeI/EcoRI-cleaved pET28a. The resulting plasmid (pET28_ *fasR*$_H$) was transformed into BL21 λ (DE3) cells. The plasmids to generate the FasR$_{\Delta 33}$ and FasR point mutants (FasR$_{LVL}$, FasR$_{L106F}$, FasR$_{L98A}$ and FasR$_{F123A}$) as recombinant proteins with a hexa-histidine-TEV-tagged site at the N-terminus were synthesised

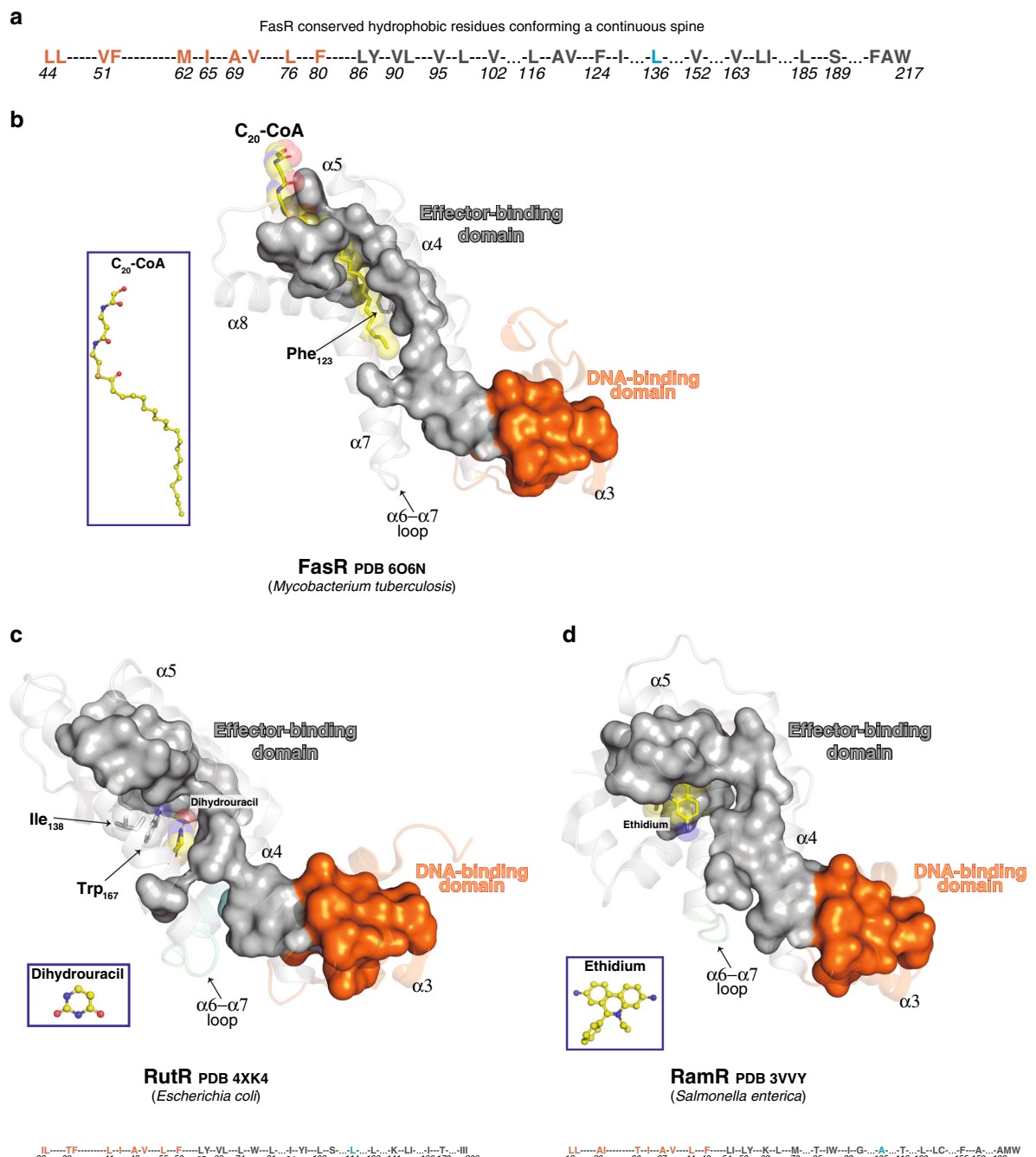

**Fig. 6 A hydrophobic spine connects input effector-sensing to output DNA-binding. a** Hydrophobic-spine amino acids in FasR. The colour code depicts residues of the DNA-binding domain in orange, the effector-binding domain in grey and the α6–α7 loop in cyan. **b–d** distinct TFRs (FasR, RutR, RamR) illustrate the conservation of the hydrophobic spine (residues shown as molecular surface), connecting effector- to DNA-binding domains. The colour code is identical to panel **a**. The effector molecules in atom-coloured sticks are labelled, with transparent spheres overlaid (insets show their markedly disparate structures). See Supplementary Fig. 13 for additional examples.

(GenScript). NdeI/HindIII restriction sites were engineered in a pUC57-Am plasmid where genes of interest were introduced. Synthetic plasmids were digested with NdeI and HindIII, extracted from the agarose gel, and inserted into the expression vector pET28a, as described by the manufacturer (Merck). The recombinant plasmids were transformed into BL21 CodonPlus(DE3)-RIL cells and transformants selected on LB agar plates with 50 μg/ml kanamycin. DNA sequences of all genes were verified by Sanger sequencing.

**Expression and purification of proteins.** Expressions of all N-terminally hexa-histidine-TEV-tagged or hexa-histidine-tagged proteins used in this work were carried out following isopropyl-β-thiogalactoside (IPTG) induction in BL21 CodonPlus(DE3)-RIL or BL21 λ (DE3) E. coli. Bacteria were grown at 37 °C in

500 ml LB broth to 0.6–0.7 absorbance at 600 nm. IPTG was then added to 0.3–0.5 mM and the culture was grown for 12 h at 23 °C. Cells were harvested by centrifugation at 2800 × g at 4 °C, resuspended in 30 ml of lysis buffer (50 mM Tris. HCl pH 8, 150 mM NaCl, 5 mM imidazole, 10% glycerol, 10 mM β-mercap-toethanol) and lysed by sonication. After centrifugation (25,000 × g, 30 min, 4 °C), the supernatant was recovered and FasR, FasR$_{Δ33}$ or FasR point mutants were separated from whole-cell lysates by Ni-NTA agarose chromatography (Qiagen, Inc). After three washing steps with lysis buffer, His$_6$-tagged FasR were eluted from the resin with 250 mM imidazole in lysis buffer, dialysed overnight against FasR Buffer 1 (10 mM Tris.HCl pH 8, 300 mM NaCl). In the case of hexa-histidine-TEV-tagged proteins (FasR, FasR$_{Δ33}$, FasR$_{LVL}$, FasR$_{L1069F}$, FasR$_{L98A}$ and FasR$_{F123A}$) after three washing steps with lysis buffer, 0.5 mg TEV protease and DTT to 1 mM final concentration were added. The mixture was incubated 2–3 h at 23 °C and 12 h

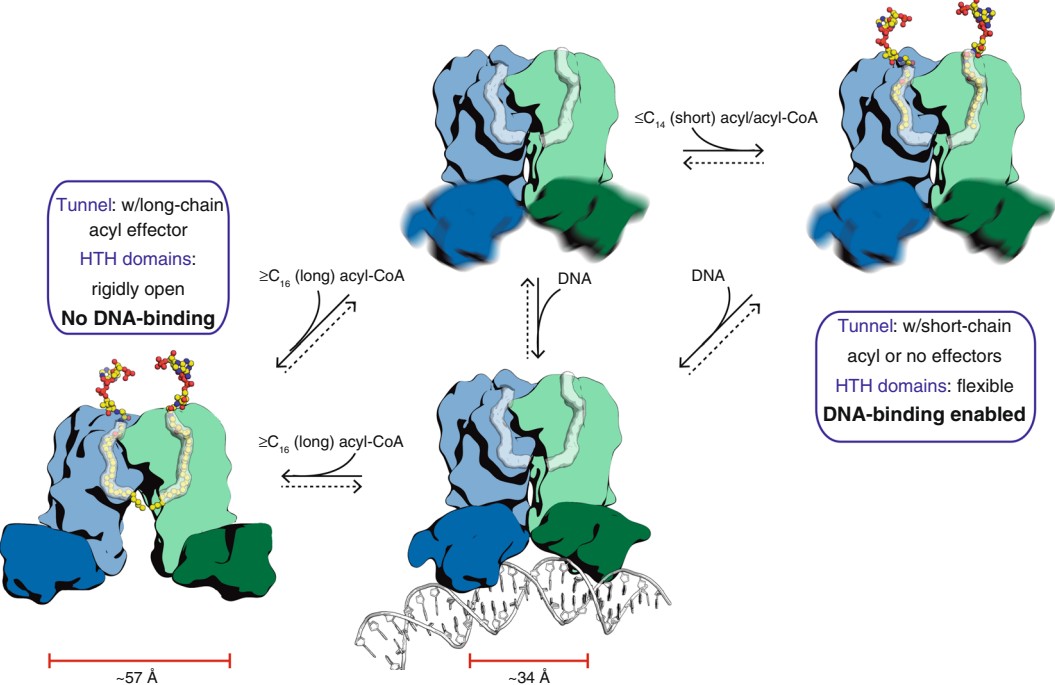

**Fig. 7 FasR-mediated long acyl-CoA sensing and response mechanism.** The model combines information from the three crystal structures presented in this report (FasR$_{\Delta 33}$-C$_{14}$, FasR$_{\Delta 33}$-C$_{20}$-CoA and FasR–DNA complexes). A free structure of FasR with no bound effectors nor DNA has not been determined experimentally, and is not currently known whether it builds up to detectable concentrations within the living cell. The dotted arrows reflect plausible equilibria. See Supplementary Movie 3 to better grasp the anticipated dynamics.

at 4 °C. Proteins were eluted from the resin with 5 mM imidazole in lysis buffer and dialysed overnight against FasR Buffer 2 (10 mM Tris.HCl pH 8, 300 mM NaCl, 5% glycerol). A final size-exclusion chromatography step (S200-Superdex, GE) was performed for all proteins, pre-equilibrating the column in FasR Buffer 2 and eluting isocratically at 0.5 ml/min with high-performance liquid chromatography (Akta Purifier, GE). FasR$_{LVL}$ was the only mutant that showed detectable levels of monomeric form eluting from SEC (all other proteins eluting as dimeric species), in which case the peak corresponding to the dimer was recovered for further functional analyses by EMSA. Protein purity was controlled by Coomassie blue staining after SDS-PAGE on a 15% polyacrylamide gel. Protein concentrations were determined by UV spectroscopy. Purified proteins were stored at 4 °C.

**Surface plasmon resonance**. Surface plasmon resonance analyses were performed with a Biacore T100 instrument (GE Healthcare). Pure FasR and FasR$_{F123A}$ were dialysed against 10 mM sodium acetate pH 4.5 and coupled to a CM5 sensor chip using the Amine Coupling Kit (GE Healthcare). Micromolar concentrations of C$_{20}$-CoA were dialysed against 25 mM Tris.HCl pH 8, 150 mM NaCl, 0.005% Tween 20. Serial twofold dilutions were made in the same buffer and injected over the chip surface. Dissociation was then carried out by injecting buffer alone. Nonspecific binding was considered by injecting identical analyte concentrations over a control surface with no protein. Experiments were performed at 25 °C by triplicate, producing standard deviations of <12%. Data were analysed using Biacore T100 Evaluation software.

**Electrophoretic mobility shift assays (EMSAs)**. His$_6$-tagged FasR and FasR point mutant proteins (FasR$_{LVL}$, FasR$_{L106F}$, FasR$_{L98A}$ and FasR$_{F123A}$) were used to assess the protein binding to P*fas*$_{MT}$ (398 bp) promoter fragment. The promoter DNA fragment for these assays was generated by PCR amplification from *Mtb* genomic DNA with primers N2_Fas1Mt-prom (5′-CATAACGATTTGATAACAAAACT GC-3′) and C_Fas1Mt-prom (5′-CACCCGGTCGTGCTCGTGGATCGTC-3′). N2_Fas1Mt-prom primer was end-labelled with [γ-$^{32}$P] ATP (3000 Ci mmol$^{-1}$) using T4 polynucleotide kinase and the PCR product obtained was purified from agarose gels. Proteins were either pre-incubated or not with acyl-CoAs and then the mixtures incubated with $^{32}$P-labelled probe (1000–5000 cpm) in a total volume of 25 μL binding buffer (25 mM Tris.HCl pH 8, 1 mM PMSF, 5% (v/v) glycerol, 5 mM MgCl$_2$, 150 mM NaCl and 1 μg poly-dIdC) at room temperature for 15 min. DNA-protein complexes were resolved by electrophoresis on a 6% (w/v) non-denaturing polyacrylamide gel in 1X TBE (89 mM Tris base; 89 mM boric acid; 2 mM EDTA), 5% (v/v) glycerol at 150 V, on ice. Results were visualized and recorded with a Typhoon™ FLA 7000 scanner (GE).

The equilibrium dissociation constant ($K_D$) is a quantitative measurement to assess the affinity of biological interactions. For the FasR:DNA EMSA binding experiments described in this work we define the $K_D$ as the concentration of FasR for which 50% of the DNA is in complex with the protein. The relationship between $K_D$ and affinity is reciprocal, lower $K_D$s correspond to higher affinities. After performing binding reactions in which the protein is titrated, the fraction of DNA bound at each concentration of protein is calculated and the data are adjusted to a binding equation using non-linear regression[41]. Densitometry of DNA bands was quantified as well as the maximal fraction bound ($B_{max}$) plateau. The fraction of bound DNA was plotted as a function of protein concentration and fitted to Eq. (1) using GraphPad Prism software to perform non-linear regression:

$$\text{Fraction bound} = B_{max} * [P]/(^{\text{app}}K_D + [P]) \tag{1}$$

Knowing $[P]$ = protein concentration, the apparent dissociation constant ($^{\text{app}}K_D$) can be quantified as well as the maximal fraction bound ($B_{max}$).

Acyl-CoA effectors trigger FasR:DNA dissociation, and can thus be treated as non-competitive binder inhibitors. Dose–response experiments were performed to assess inhibition activity, using a modified EMSA protocol. A constant and saturating concentration of protein was equilibrated with 0.3 nM labelled DNA and increasing concentrations of acyl-CoAs in equilibration buffer. The EMSA data were collected as above and fitted to the sigmoidal dose response Eq. (2) in GraphPad Prism to determine the acyl-CoA concentration required to displace half of the bound FasR:DNA complex (IC$_{50}$):

$$\text{Fraction bound} = 100/\left(1 + 100 \times \left(E_f - \log \text{IC}_{50}\right)\right) \tag{2}$$

$E_f$ = logarithm of acyl-CoA concentration.

EMSA experiments with FasR$_{wt}$ and selected FasR mutants were performed by triplicate, to express $^{\text{app}}K_D$ and IC$_{50}$ as average values ± one standard error of the mean.

Apparent inhibition constants ($^{\text{app}}K_i$) were also calculated[42,43], to correct IC$_{50}$ figures taking into account the $K_D$ as well as the concentrations of labelled DNA ($[D]$) and protein ($[P]$) according to the Lin and Riggs conversion Eq. (3):

$$^{\text{app}}K_i = 2 * K_D * \text{IC}_{50}/(2[P] - [D] - 2K_D) \tag{3}$$

**Controlled proteolysis**. For pre-incubation, 2 mM C$_{20}$-CoA was mixed with 0.2 mM wild-type His$_6$-tagged FasR in 10 mM Tris.HCl pH 8, 0.3 M NaCl for 1 h at 25 °C. FasR (16 μM), either pre-incubated or not with C$_{20}$-CoA, was incubated with trypsin (32 nM; Promega, V511C) at 37 °C in the same buffer. At different time points, aliquots were drawn, the digestion stopped by adding SDS sample buffer

and immediately boiled and analysed by SDS-PAGE. Gels were stained with Coomassie Brilliant Blue, digitalized with a Typhoon™ FLA 7000 scanner (GE) and densitometry quantification of the full-length band was performed with ImageJ. Observed degradation was fitted with an exponential function to compare decay rates.

**Crystallisation and data collection.** FasR$_{\Delta33}$ (5 mg/ml) crystallised at 20 °C, mixing 2 + 2 µl of protein and reservoir solution (0.1 M MES monohydrate pH 6.0, 22% v/v polyethylene glycol 400) using 1 ml reservoir on a hanging-drop vapour-diffusion setup. Transferred to mother liquor with 20% (v/v) glycerol as cryo-protectant, crystals were mounted in cryo-loops (Hampton Research), and flash cooled in liquid nitrogen (see Supplementary Notes for extended details and references). X-ray diffraction data were collected at −163 °C at the in-house Protein Crystallography Facility of the Institut Pasteur de Montevideo (Uruguay) with a MicroMax-007 HF rotating Cu anode (Rigaku) and a Mar345 image plate (marXperts) controlled with the proprietary mar345dtb software.

FasR$_{\Delta33}$ (5 mg/ml) was crystallised in complex with C$_{20}$-CoA using a 1:1 molar stoichiometry of the acyl-CoA ligand in the crystallisation drops (2 µl protein:ligand + 2 µl mother liquor 2.2 M NaCl, 0.1 M Na-acetate trihydrate pH 4.7). A hanging-drop vapour-diffusion setup was incubated at 20 °C using 1 ml mother liquor as reservoir. Crystals were transferred to mother liquor with 25% (v/v) glycerol, mounted in cryo-loops and flash cooled in liquid N$_2$. X-ray diffraction data were collected at −173 °C at SOLEIL synchrotron (PROXIMA 1 beamline, France), using a PILATUS 6 M detector (Dectris) controlled with the open source MXCube software.

FasR–DNA crystals were obtained by co-crystallisation. Double-stranded DNA was generated by co-incubating the two complementary oligonucleotides FwPfas25nt (5′-TACCCGTACGTAGAACTCGCCAGTA-3′) and RvPfas25nt (5′-TACTGGCGAGTTCTACGTACGGGTA-3′) under standard slow-cooling hybridisation conditions. Double-stranded DNA was mixed in a 1:1 molar stoichiometry with full-length FasR (5 mg/ml) and set for hanging-drop vapour-diffusion crystallisation by mixing 2 µl of protein:DNA with equal volume of mother liquor (28% (w/v) PEG monomethyl ether 2000, 0.25 M ammonium citrate pH 7.0, 0.1 M imidazole) over 1 ml mother liquor as reservoir solution. Crystals were transferred to mother liquor with 35% (v/v) PEG 400, mounted in cryo-loops and flash cooled in liquid N$_2$. X-ray diffraction data were collected at −173 °C at Diamond Light Source synchrotron (I04-1 beamline, UK), using a PILATUS 6M detector (Dectris) controlled with the open source Generic Data Acquisition software.

Bragg diffraction intensities were integrated with XDS[44], and scaled and reduced to amplitudes with Aimless and Ctruncate[45].

**Structure determination and refinement.** The structure of FasR$_{\Delta33}$-C$_{20}$-CoA was solved ab initio with Arcimboldo[15] which uses Phaser[46] as molecular replacement (MR) engine to place α-helices, and ShelxE[47] for density modification and chain-trace extension. The structure of FasR$_{\Delta33}$-C$_{14}$ was solved by MR[47] using the refined FasR$_{\Delta33}$-C$_{20}$-CoA model as search probe. Buster[48] was used to refine both FasR$_{\Delta33}$-C$_{20}$-CoA and FasR$_{\Delta33}$-C$_{14}$ atomic models, iterating with manual model rebuilding and validation with Coot[49]. Final validation was done with MolProbity[50]. OMIT maps were calculated for the FasR$_{\Delta33}$-C$_{20}$-CoA and FasR$_{\Delta33}$-C$_{14}$ structures using final refined models from which only the acyl-CoA ligands were omitted to calculate structure factors (see Supplementary Fig. 2 for further details). Final refined maps were also used to calculate real-space correlation coefficients with respect to model-derived maps, on a per residue basis[51].

The FasR–DNA complex was solved by MR[46] using a portion (corresponding to the dimeric regulatory domains with no ligands nor HTH domains included) of the refined FasR$_{\Delta33}$-C$_{20}$-CoA model as search probe. Initial refinement[52] with this partial model reduced R-factors to <50%, and produced Fourier difference maps that clearly revealed the presence of both HTH domains and double-stranded DNA. Limited resolution and model incompleteness resulted, however, in mediocre mF$_{obs}$-DF$_{calc}$ maps at this point. ShelxE[47] was instrumental at improving electron density continuity, using the unrefined MR solution model and diffraction intensities as inputs, a larger than usual sphere of influence (5 Å) for density modification, the free-lunch option set at 3.85 Å to better handle data incompleteness at the higher resolution shells, and testing different solvent contents (from 0.4 to 0.6). All output maps were visualized superposed, readily allowing for manual main-chain tracing[49] of HTH and DNA portions, only including residues clearly visible in electron density at each cycle. This procedure was cycled iteratively using the progressively more complete protein models, running 5 cycles of ShelxE density modification each time. For the manual model (re)building, recent developments for real-space fitting in very low-resolution maps within Coot[53] proved essential, using Prosmart-generated external restrains, map blurring and optimized Geman-McClure parameters (alpha = 0.4 proved best) for optimal modeling of DNA within electron density.

The orthorhombic C222$_1$ space group of the FasR–DNA crystals was confirmed by several standard procedures[45,54], notably including integration in the corresponding triclinic group and using molecular replacement as a means of deducing real symmetry. However, the DNA molecule (one double-stranded oligonucleotide per ASU) was found sitting with its long axis approximately perpendicular to one of the orthorhombic twofold axes, revealing both alternative

5′→3′ orientations to be present in different unit cells of the crystal. This form of static disorder is frequent when a crystal contains two or more very similar molecules (such as complementary strands in DNA with palindromic sequences), which may occupy the same lattice position with low occupancy. Refinement was performed with phenix.refine[52], using external restrains from the high resolution FasR models described above and overall strategies to deal with low-resolution data. Intermediate cycles of real-space refinement[55] were also instrumental. The occupancies of DNA atoms were reduced to 0.5, avoiding bumping restraints.

Difference Fourier maps very clearly revealed the positions of the two missing DNA-binding domains as well as the double-stranded DNA molecule early on during refinement. However, difference maps did not show any detectable signal at the expected positions of putative ligands bound within the effector-binding tunnel. To confirm this, the acyl moieties from the FasR$_{\Delta33}$-C$_{14}$ model were superimposed in place within the FasR–DNA effector-binding tunnel, manually regularised to minimise clashes and added into the FasR–DNA model for further restrained refinement. Not only the crystallographic R-factors increased significantly (for both working and free sets of reflections), but difference Fourier maps revealed >5 σ negative peaks on the acyl atoms, indicating they are not actually present in the crystal.

Structural analyses were done with the CCP4 suite[56], PISA[57] and illustrations produced with Pymol[58].

**Structural bioinformatics to define the hydrophobic spine.** FasR structural homologues were searched with PDBeFold[59], thus retrieving a wide range of sequence similarities. Structural alignment of such hits was performed with T-Coffee Expresso[60]. The resulting 337 sequences were filtered keeping only 76 that had <80% identity. This multiple sequence alignment (MSA) served to generate a hidden Markov model (HMM) profile using the hmmbuild routine[61] within the program suite HMMER. The UNIPROT database was searched with this HMM profile using the hmmsearch module in HMMER, and then CD-HIT[62] was used to filter redundant sequences, ultimately producing a list of 2591 sequences, all containing TFR effector-binding and DNA-binding domains. A MSA was calculated with T-Coffee in M-Coffee mode[63], and the resulting alignment (Supplementary Data 1) allowed the calculation of observed frequencies for each one of the 20 amino acids for each position in the MSA. A score for each MSA position was then generated by multiplying each frequency by the corresponding hydrophobicity index[64] and summing for all 20 amino acids (Supplementary Data 2). The hydrophobic spine was defined as the set of positions with a weighted hydrophobicity index (which goes from −4.5 [hydrophilic] to 4.5 [hydrophobic]) equal or larger than 2. A comparison with three additional hydrophobicity index scales, based on different criteria, resulted in essentially identical results (Supplementary Data 2). Sequence alignment figures were prepared with Esprit[65].

**Molecular dynamics simulations.** The FasR$_{\Delta33}$-C$_{26}$-CoA complex was built using the FasR$_{\Delta33}$-C$_{20}$-CoA (PDB 6O6N) model as template. The bound acyl-CoA was manually extended by six carbons using Pymol[58]. C$_{26}$-CoA was optimized and 10,000 rotamers were generated with RDKit (http://www.rdkit.org). Energy minimisation was performed using the Rosetta suite[66] using dimer symmetry constraints, harmonic restraints were used to preserve the ligands positions as observed in the crystal structure, and 10,000 models were generated. The best complex was selected based on Rosetta energy score, optimal stereochemical geometry and no clashes. The selected model was used as starting structure for classical molecular dynamics simulations using Gromacs 2018_cuda8.0 and GROMOS96 43a1 force field[67]. An octahedron box was solvated and charge-balancing counterions were included to neutralise charges[68]. Initially, the system was relaxed by energy minimisation, and then equilibrated for 200 ps using a reference temperature of 27 °C. Simulations were performed for 10 ns with no constraints, recording snapshots every 5 ps for analysis. Identical strategy was repeated for the protein alone (the effector ligand was removed from the starting model) and root-mean squared deviation in atom positions were calculated for different domains as defined in the text and figure captions.

**Reporting summary.** Further information on research design is available in the Nature Research Reporting Summary linked to this article.

## Data availability

Data supporting the findings of this manuscript are available from the corresponding authors upon reasonable request. A Reporting Summary for this Article is available as a Supplementary Information file. Source data are provided with this paper. Macromolecular 3D structural data (model coordinates and crystallographic structure factors) presented in this study have been deposited in the wwPDB with accession codes PDB 6O6O, PDB 6O6N and PDB 6O6P. Raw X-ray diffraction data for each one of those structures have been deposited in SBGrid with Digital Object Identifiers 10.15785/SBGRID/648, 10.15785/SBGRID/647 and 10.15785/SBGRID/649, respectively.

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

## Acknowledgements

We thank Stanislas Leibler, Michael Mitchell and Pablo Sartori for sharing initial strain analysis scripts; Matias Machado for assistance in molecular dynamics; Frank Lehmann for initial cloning efforts; Sebastian Klinke (Fundación Leloir) and the staff at Proxima 1 beamline (Soleil synchrotron) and at I04-1 beamline (Diamond synchrotron) for assistance with data collection. We acknowledge computational and storage services (TARS cluster) provided by the Institut Pasteur IT Dept (Paris). We thank the CCP4/CeBEM Macromolecular Crystallography School (USP@São Carlos, 2018), especially Isabel Usón and Paul Emsley for helping us, respectively, with ShelxE and Coot, in dealing with low-resolution density modification and model building. J.L. traineeships at IPasteur-Montevideo were funded by CeBEM (www.cebem-lat.org). Support to A.B. from Institut Pasteur (grant 761-International_Joint_Research_Unit-IMiZA-2016), to G.G. from ANPCyT (grant PICT 2015-0796) and to H.G. from ANPCyT (grants PICT 2012-0168 and 2022) and NIH (grant 1R01AI095183-01) are acknowledged.

## Author contributions

J.L., L.D., N.L., M.F. and F.T. carried out the experiments and acquired the data; J.L., L.D., G.G., F.T., E.M., H.G. and A.B. conceived the study and performed data analyses and interpretation; J.L., H.G. and A.B. wrote the paper; H.G. and A.B. substantively revised the paper and coordinated the project. All authors gave final approval for publication.

## Competing interests

The authors declare no competing interests.
