## [Peer Review File · Nature Communications]

Peer Review File – Reviewers comments, first round

Reviewers' comments:

Reviewer #1 (Remarks to the Author):

The manuscript by Lara et al presents a structural analysis of the conformation of the ligand-binding pocket in the acyl-CoA responsive FasR transcription factor in *M. tuberculosis* and of the allosteric mechanism underlying the inhibition of DNA binding upon ligand interaction. Three crystal structures are presented, two ligand-bound and one DNA-bound. A detailed analysis of the acyl-CoA binding tunnel guides the design of site-directed FasR mutants, which are constructed and are analyzed for ligand-induced inhibition of DNA binding using electrophoretic mobility shift assays. Finally, a generalized model is presented of the allosteric output upon ligand binding for TetR-family transcription factors. The manuscript is well-written and high-quality figures and movies are presented. It is a well-executed and detailed structural work, with the potential to guide novel drug design approaches, however the methodology of the work is mainly limited to protein crystallography and novel insights related to biological relevance are lacking. Although the specific architecture of the ligand-binding pocket - a long hydrophobic tunnel crossing the entire effector binding domain (enabling (differential) response to long-chain acyl-CoA molecules) - appears novel, this is not the case for the mechanistic model. The observation that ligand binding “opens up” the DNA-binding domains resulting in a suboptimal spacing for DNA binding has been made before for a variety of TetR transcription factors, including other acyl-CoA-responsive TetR regulators.

1. One of the site-directed mutants, FasRLVL, exists as a heterogeneous population of monomers and dimers (Suppl. Figure 2). Nevertheless, a similar molar amount of this mutant protein resulted in the formation of FasR-DNA complexes as the WT protein (Figure 3b). Could the authors comment on this observation? Have attempts been made to separate the monomeric from the dimeric population? Are monomers capable of binding DNA with a similar affinity as dimers?

2. The functional analysis of the mutant proteins is quite limited, testing only a single protein concentration and a limited number acyl-CoA conditions - a statistical analysis is missing. For example, the conclusion with regards to the L98A mutant (“both mutant variants showed however significant functional effects in uncoupling ligand-binding with DNA-association (Fig. 3d),...”) is difficult to follow when assessing the EMSA result presented in Fig. 3d. It is advised that the authors extend this analysis by testing wider concentration ranges, performing replicate experiments and performing a quantitative analysis (either with an EMSA or with a SPR approach). For each mutant, DNA binding could be assessed quantitatively with a K_D calculation and ligand-induced inhibition of DNA binding could be assessed with the calculation of an inhibition constant (K_i). SPR could also be used to test the difference in affinity between short- and long-chain acyl-CoAs.

3. In the mechanistic model for acyl-CoA sensing and response presented in Figure 6d, the authors hypothesize that FasR is capable of existing without associated acyl-CoA effector or DNA. In the legend of Figure 6 it is mentioned “the free structure of FasR with no bound effectors nor DNA, has not been determined yet.” Can the authors provide arguments that such an apo-state of the protein exists (given that crystallization of the protein after heterologous expression in *E. coli* resulted in an acyl-CoA-bound form)?

Reviewer #2 (Remarks to the Author):

General comment:

The manuscript presents the structural and functional characterization of the *Mycobacterium tuberculosis* FasR (Rv3208), a transcriptional factor that controls in particular the expression of the fatty acid synthase (*fas*) gene. FasR protein belongs to the well-known family of TetR regulators, containing a DNA-binding module HTH and a dimerization domain which is also the ligand/effector binding domain. The crystal structures of FasR have been solved in three different states, in complex with the effector fatty acid C20-CoA, in complex with its DNA operator and in unliganded (apo) form. However, the latter form turns out to contain in the ligand binding pocket a fortuitous ligand, presumably a hydrophobic molecule that was modeled as a C14 alkyl chain. Based on structural analysis, four different FasR mutants were produced and studied for their ability to bind DNA in the presence or in the absence of the effectors C16-CoA and C20-CoA. A molecular dynamics simulation was further performed showing that the FasR structure can accommodate fatty acyl C26-CoA. From obtained data and analysis, the authors propose a functional mechanism of action for the FasR: the binding of long fatty acyl-CoA induces an open rigidified conformation, incompatible with binding to a double stranded DNA, whereas the apo form (or even the form in complex with a short fatty acyl) is a closed conformation, with higher flexibility of the HTH domains allowing the proper binding to DNA. Finally, the existence of a hydrophobic spine, conserved in all TetR-like transcription factors and playing a role in allosteric movements, was highlighted.

This study is a good piece of work, interesting and well conducted. However, the obtained structures and the described functional mechanism of FasR are not particularly original but rather very common in TetR regulators. Furthermore, some experiments seem missing to reinforce the results and the discussion appears not strongly supported by available data, but rather speculative. Finally, there are a number of issues that in my opinion affect the authors' main conclusions and should be considered.

Specific comments:

- The authors claim the essentiality of FasR (Rv3208). However, I think it is still controversial. FasR is listed as a non-essential gene for in vitro growth of H37Rv by analysis of saturated Himar1 transposon libraries (DeJesus et al., 2017), whereas the previous similar study of Griffin et al. (2011) had found FasR as essential.
- Do the authors have an idea of why the transcriptomic study of Rustad et al (Genome Biology, 2014, 15:502) does not corroborate the functional role of FasR, i.e. an overexpression of *fasR* (+2.33 in log2) in *M. tuberculosis* induced only a weak effect on *fas* and *acpS* expressions (Rv2524c: +0.59; Rv2523c: -0.64) ?
- The references cited in the first page of the introduction should be controlled, i.e. ref #5 before the #4 and the reference #3 do not appear to be the appropriate reference.
- According the structure validation report for FasR/C20-CoA, the ligands (C20-CoA and C14) do not fit perfectly well in the electron densities. Could the authors show omit maps for ligands as supplementary figures, and also add in Table 1 for ligands some local density-based validation

metrics, such as the real-space R-factor, the real-space correlation coefficient and/or the real-space Z-difference/observed. These additional data could be discussed in the text.

- Apparently, all constructions of the four mutants unfortunately contain the N-terminal part. It would have been great to solve the crystal structures of the “blocked” mutants (i.e. L106F and (L106F, V163F, L185F)), hoping to obtain an apo form and observing the effects on HTH organization and flexibility.

- EMSA results with the mutant L98A are not very convincing and very similar to those with wild-type protein. Do you have an explanation?

- Why only the mutant F123A has been tested by SPR? It would have been interesting to see if the mutant L98A also conserves the capacity to bind C20-CoA as the wild-type. In the same way, SPR should be used to further confirm by another experimental technique that the two “blocked” mutants have lost their capacity to bind to C20-CoA and therefore cannot be modulated in their DNA-binding activity (i.e. confirming the Figure 3b).

- I did not understand very well the interpretation of the mutant F123A. This mutant can accommodate C20-CoA while retaining its ability to bind DNA. I cannot imagine that this mutated protein can bind at the same time the C20-CoA effector molecule and the DNA. Surely, further experiments should be realized to understand this mutant (as for instance by SPR or X-ray).

- Can the authors explicitly confirm in the text that no constraints were applied during the 10ns molecular dynamics simulation?

- I am not an expert in molecular dynamics simulation. However, it seems to me that the reliability of results should be assessed by comparing several MDS results, like “positive” and “negative” controls (for example, as positive control, an identical MDS with the C20-CoA and as negative control, may be a PEG400 molecule).

- To experimentally validate the MDS, SPR with the C26-CoA could be performed.

- Please give the observed distance between HTH in the DNA-FasR complex structure.

- In discussion, the authors stated the volume delimited by the two protomers is unusually large in FasR. Compared to which tetR structures? Did they perform a survey of all solved TetR structures? Can we have some numerical values?

- In discussion, may be the following sentence should be rewritten “(in FasR-C14 complex) significantly higher flexibility of the HTH domains, as revealed by higher atomic displacement parameters and weaker electron density”. I think the HTH domains were well defined in the electron density whereas connecting parts between the HTH and the effector binding domains are more flexible and not well defined.

- Deciphering how the allosteric signal in tetR family of regulators is transmitted from the effector binding domain to the HTH module remains highly challenging. To the best of my knowledge, nobody has yet demonstrated how this phenomenon happens. The authors suggest the existence of a hydrophobic spine common to TetR regulators and participating to the allosteric signal

transduction upon effector binding. I am not really convinced by the transmission spine proposition. I do not think the results presented in the manuscript allow to extrapolate a general allosteric mechanism for all members of TetR family. Furthermore, the residues of the hydrophobic spine do not appear conserved across members of TetR family, according the supplementary Figure 1 and according images of Figures 6a, 6b 6c and the Supplementary Figure 5. It is mentioned that “a comprehensive MSA that confirms strict conservation id not shown”. This MSA should be provided to convince readers. There is clearly a lack of evidence precise and consistent enough to support the transmission spine hypothesis.

Reviewer #3 (Remarks to the Author):

This is a very nice paper investigating the function and mechanism of a key mycobacterial detail using a wide range of biochemical, biophysical and structural methods. Based on the results the authors propose a very plausible model for activation which will have wider implication on the field of bacterial repressors. Tuberculosis remains a major global challenge and hence basic research in academia will be vital to develop new and better therapies. The paper is therefor well suited for a more general journal such as Nature Communication.

Overall, the paper is well written, and of the highest technical level. The only drawback is the limited resolution of the FasR-DNA structure, however, the authors successfully avoided over-interpretation and given the well-known difficulties to obtain high-resolution crystals structures of (flexible) protein DNA complexes (which also makes these complexes almost impossible to tackle by cryoEM) this should not prevent publication. Overall, I recommend publication with minor revisions

However, there are a few points I would like the authors to consider for a revised submission:

(0) The importance of structures in the search for new TB therapies could be mentioned in the introduction, there are several reviews for example from the Mtb structural genomics consortia that could be cited.

(i) Although the authors mention the mycobacterial EthR in one of their supplementary material, I would like to see more of a comparison. EthR should for example be in the sequence alignment as it would be interesting to know if the flexible N-terminus is present as well. In addition, both members of the TetR family seem to bind similar ligands, hence a more detailed comparison of the respective binding sites would be of interest. Also, least-squares superpositions of FasR with EthR could show if the the proposed mechanism could be employed by EthR (and other members of the family).

(ii) The unbiased density of all ligands should be given in the supplementary material.

(iii) Although the FasR-DNA structure is of limited resolution, a least-squares superposition with known TetR-DNA structure (ribbon or trace) would be helpful. The resolution is not sufficient to determine if a ligand is bound or not.

(iv) Overall the materials and methods are comprehensive, in particular in the crystallographic section, a few references may be missing and could be add, ie. T.-Y. Teng for cryocooling techniques, I. Ascone (2007) for the beam line, Broennimann et al, 2006 for the Pilatus detector, and since

Arcimboldo was used on a DNA binding protein, Proepper et al, (2014) could be added. If local non-crystallographic restraints were used, Uson I., et al. (1999) could be cited. Beam line (or in-house course) and wavelength used should be added to the crystallographic table in the main text.

Lara J et al. “*Mycobacterium tuberculosis* FasR senses very long fatty acyl-CoA with a tunnel and a transmission spine”

Point-by-point response to the reviewers’ comments and suggestions.

General concepts.

Thank you very much for your feedback, the overall positive comments from the three reviewers encouraged us to put substantial additional work into preparing a resubmission. The suggested modifications and additional data have strengthened the paper in the line of its main message.

To summarize the most prominent additional data that have been produced to address the reviewers’ concerns:

- a much more thorough set of electrophoretic mobility shift assays has been performed, allowing to quantify FasR’s affinity for its cognate DNA-binding, and as importantly, quantifying acyl-CoA effectors’ association constants. These analyses compared the wild-type protein with all the different point mutants, both the variants that occlude the ligand-binding tunnel or the ones that uncouple the ligand-triggered allosteric effect. These data address several of the reviewers concerns and further the mechanistic insight of the signalling process.

- a much more comprehensive analysis of the hydrophobic spine was performed, now including over 2500 different TFR sequences. The bioinformatic algorithm ensured the inclusion of a very broad range of sequence variants.

- OMIT maps and real space correlation coefficients were calculated as requested, sontrlgy confirming the presence and accurate refinement of the C₂₀-CoA and C₁₄ effector ligands bound within the tunnel.

- A clearer discussion has been put forward, highlighting the difference between the allosteric model we are now proposing versus the previous open-closure mechanism. Our interpretation is consistent with all the data available on TFRs’ signal-triggered regulation, whereas the open-closure paradigm was inconsistent with some lines of evidence among which the finding that apo-protein structures closely resemble the ligand-bound or induced, open form, rather than the DNA-bound, closed form.

- Many other corrections have now been included, following the reviewers’ comments.

Specific responses

Reviewer #1:

General remarks

The manuscript by Lara et al presents a structural analysis of the conformation of the ligand-binding pocket in the acyl-CoA responsive FasR transcription factor in M. tuberculosis and of the allosteric mechanism underlying the inhibition of DNA binding upon ligand interaction. Three crystal structures are presented, two ligand-bound and one DNA-bound. A detailed analysis of the acyl-CoA binding tunnel guides the design of site-directed FasR mutants, which are constructed and are analyzed for ligand-induced inhibition of DNA binding using electrophoretic mobility shift assays. Finally, a generalized model is presented of the allosteric output upon ligand binding for TetR-family transcription factors. The manuscript is well-written and high-quality figures and movies are presented. It is a well-executed and detailed structural work, with the potential to guide novel drug design approaches, however the methodology of the work is mainly limited to protein crystallography and novel insights related to biological relevance are lacking. Although the specific architecture of the ligand-binding pocket - a long hydrophobic tunnel crossing the entire effector binding domain (enabling (differential) response to long-chain acyl-CoA molecules) - appears novel, this is not the case for the mechanistic model. The observation that ligand binding “opens up” the DNA-binding domains resulting in a suboptimal spacing for DNA binding has been made before for a variety of TetR transcription factors, including other acyl-CoA-responsive TetR regulators.

We thank the reviewer for the mostly positive comments about the quality of our work. Although the reviewer does appreciate the novelty of the ligand-binding architecture (highly relevant for sensing long- and very long-chain fatty acyl moieties, a key aspect for the regulatory function of FasR in mycobacteria), he/she also comments about the lack of biological relevance of this work. We believe that confirming for the first time, through structural studies, that very-long chain acyl-CoAs synthesized by the main fatty acid synthase complex, are the ligands that regulate *de novo* fatty acid biosynthesis in *M. tuberculosis* is a relevant finding with biological implications. For example, structure-based approaches have been used to find inhibitors of EthR, a TetR repressor of EthA, inhibitors that boost the bioactivation of ethionamide, a second-line antituberculosis drug that inhibits mycolic acid biosynthesis (Willand et al 2009 *Nat Med* 15:537-44). Our own recent findings confirming that FasR plays a key role in the *M. tuberculosis* infection process (to be published elsewhere), suggest that a structure-guided approach could likely be used to design FasR inhibitors. Such compounds could then be tested as single inhibitors of *M. tuberculosis*, or as effective boosters of known antimycobacterial drugs. We now elaborate this perspective more clearly in the Discussion.

Another issue raised by the reviewer concerns the effector-triggered mechanism controlling the regulator's DNA-binding function. What the reviewer understood from the data and our mechanistic model, is that our proposal corresponds to the same interpretation that has been widely formulated in the past for TetR-like regulators, hence lacking any novelty. We greatly appreciate for having raised this point, as we firmly believe this is due to a lack of clarity in the way we conveyed our message.

The mechanistic model we are supporting is not the classical open-closure rearrangement of the DNA-binding domains, allosterically linked to association-dissociation (respectively) of the effector ligand. Instead,

we are offering a better interpretation that integrates our own data and others': the apo structure is flexible, less compactly folded, whereas the effector-bound one is rigid (its fold being completed by the effector molecule itself). The identification of a hydrophobic spine connecting the effector-binding pocket to the HTH domains, is the molecular basis for such a compact, rigid fold that the effector is able to stabilize.

We have now substantially rephrased the discussion in this revised version to better describe the proposed model, and also to elaborate more explicitly about published experimental evidence (other than the data we are now reporting) that is consistent with, and better explained by the hydrophobic spine model.

Specific comments

1. One of the site-directed mutants, FasRLVL, exists as a heterogeneous population of monomers and dimers (Suppl. Figure 2). Nevertheless, a similar molar amount of this mutant protein resulted in the formation of FasR-DNA complexes as the WT protein (Figure 3b). Could the authors comment on this observation? Have attempts been made to separate the monomeric from the dimeric population? Are monomers capable of binding DNA with a similar affinity as dimers?

The FasRLVL mutant indeed exhibits a tendency to monomerize. Size exclusion chromatography (SEC, Supplementary Fig. 2) actually shows the separation of both populations, probably corresponding to a dimer \rightleftharpoons monomer equilibrium. Therefore, the most likely scenario is that the dimer, by binding to DNA (the DNA pseudo-palindromic box implies the association of two HTH domains), will shift the equilibrium towards the dimeric form. We don't think that the FasR monomer would be able to bind DNA with similar affinity as the dimer, but we haven't performed quantitative comparisons. Furthermore, the FasR-DNA crystal structure is also a direct evidence that the dimer is the main functional species (as in all TetR-like transcription factors so far).

We now realize that in the previous version of the ms we had not explained that for this mutant, the EMSAs were carried out with the SEC-purified dimeric fraction. We have now duly explained this in Methods. However, it must be considered that a clean experiment with the monomeric, or the dimeric forms of this mutant, might be impossible due to dimer \rightleftharpoons monomer equilibrium.

2. The functional analysis of the mutant proteins is quite limited, testing only a single protein concentration and a limited number acyl-CoA conditions - a statistical analysis is missing. For example, the conclusion with regards to the L98A mutant ("both mutant variants showed however significant functional effects in uncoupling ligand-binding with DNA-association (Fig. 3d),...") is difficult to follow when assessing the EMSA result presented in Fig. 3d. It is advised that the authors extend this analysis by testing wider concentration ranges, performing replicate experiments and performing a quantitative analysis (either with an EMSA or with a SPR approach). For each mutant, DNA binding could be assessed quantitatively with a KD calculation and ligand-induced inhibition of DNA binding could be assessed with the calculation of an inhibition constant (Ki). SPR could also be used to test the difference in affinity between short- and long-chain acyl-CoAs.

These comments are pertinent and most welcome. We have now performed additional experiments in the direction the reviewer suggested. Wider concentration ranges have been used for the extended EMSAs, replicas have now been done (so that standard errors of the mean can be reported), and the extended EMSA

results now allowed us to perform quantitative analyses (apparent K_D and K_i values), which have been incorporated in the main text of the ms.

The comparison with short-chain fatty acyls has been the point of a previous paper (Mondino et al. *Mol Microbiol* 2013 89:372). We have included now further analyses using C_{16} -CoA to be compared with C_{20} -CoA (new Supplementary Fig. 7), lending further support to the proposed hypothesis. Namely, confirming that full occupancy of the effector-binding tunnel is critical to trigger allosteric control of output DNA-binding.

3. In the mechanistic model for acyl-CoA sensing and response presented in Figure 6d, the authors hypothesize that FasR is capable of existing without associated acyl-CoA effector or DNA. In the legend of Figure 6 it is mentioned “the free structure of FasR with no bound effectors nor DNA, has not been determined yet.” Can the authors provide arguments that such an apo-state of the protein exists (given that crystallization of the protein after heterologous expression in E. coli resulted in an acyl-CoA-bound form)?

The reviewer is right. By introducing this in the conceptual model (new Fig. 7), we were implying a statement for which we don't have evidence, we don't know whether this apo state of FasR exists or not within the cell. We have rephrased the legend of the figure so that this is now made clear.

Reviewer #2:

General remarks

The manuscript presents the structural and functional characterization of the Mycobacterium tuberculosis FasR (Rv3208), a transcriptional factor that controls in particular the expression of the fatty acid synthase (fas) gene. FasR protein belongs to the well-known family of TetR regulators, containing a DNA-binding module HTH and a dimerization domain which is also the ligand/effector binding domain. The crystal structures of FasR have been solved in three different states, in complex with the effector fatty acid C20-CoA, in complex with its DNA operator and in unliganded (apo) form. However, the latter form turns out to contain in the ligand binding pocket a fortuitous ligand, presumably a hydrophobic molecule that was modeled as a C14 alkyl chain. Based on structural analysis, four different FasR mutants were produced and studied for their ability to bind DNA in the presence or in the absence of the effectors C16-CoA and C20-CoA. A molecular dynamics simulation was further performed showing that the FasR structure can accommodate fatty acyl C26-CoA. From obtained data and analysis, the authors propose a functional mechanism of action for the FasR: the binding of long fatty acyl-CoA induces an open rigidified conformation, incompatible with binding to a double stranded DNA, whereas the apo form (or even the form in complex with a short fatty acyl) is a closed conformation, with higher flexibility of the HTH domains allowing the proper binding to DNA. Finally, the existence of a hydrophobic spine, conserved in all TetR-like transcription factors and playing a role in allosteric movements, was highlighted.

This study is a good piece of work, interesting and well conducted. However, the obtained structures and the described functional mechanism of FasR are not particularly original but rather very common in TetR regulators. Furthermore, some experiments seem missing to reinforce the results and the discussion appears not strongly supported by available data, but rather speculative. Finally, there are a number of issues that in my opinion affect the authors' main conclusions and should be considered.

We thank this reviewer for considering our ms positively.

As for the lack of novelty critique, and precisely in the same line as the response we gave to reviewer #1, the reviewer understood from our data and model proposal, that this corresponds to the same interpretation that has been widely formulated in the past for TetR-like regulators.

We appreciate for having marked this point, we firmly believe this is due to a lack of clarity in the way we conveyed our message. The mechanistic model we are supporting is not the classical open-closure rearrangement of the DNA-binding domains, allosterically linked to association-dissociation (respectively) of the effector ligand. Instead, we are offering a better interpretation that integrates our own data and others': the apo structure is flexible, less compactly folded, whereas the effector-bound one is rigid (its fold being completed by the effector itself). The identification of a hydrophobic spine connecting the effector-binding pocket to the HTH domains, is the molecular basis for such a compact, rigid fold that the effector is able to stabilize.

We have now substantially rephrased the discussion in this revised version to better describe the proposed model, and also to elaborate more explicitly about published experimental evidence (other than the data we are now reporting) that supports the hydrophobic spine model.

Specific comments

- The authors claim the essentiality of FasR (Rv3208). However, I think it is still controversial. FasR is listed as a non-essential gene for *in vitro* growth of H37Rv by analysis of saturated Himar1 transposon libraries (DeJesus et al., 2017), whereas the previous similar study of Griffin et al. (2011) had found FasR as essential.

This comment is correct, the essentiality of FasR in *M. tuberculosis* is still an issue of debate. This has now been more clearly stated in the revised manuscript. We want to add here for the sake of completeness, that we have very recently demonstrated that disruption of the *fasR* gene in *M. tuberculosis* was not lethal for *in vitro* growth in a complete 7H9 medium (7H9 + OADC), but it did lead to a complete rearrangement of the lipid composition of the envelope. Furthermore, this *fasR*- mutant strain was less virulent in a mouse model and showed impaired replication in macrophages, confirming that FasR plays a critical role during *M. tuberculosis* infection. An article reporting such experimental evidence concerning the *fasR*- mutant of *M. tuberculosis* will shortly be published elsewhere.

- Do the authors have an idea of why the transcriptomic study of Rustad et al (Genome Biology, 2014, 15:502) does not corroborate the functional role of FasR, i.e. an overexpression of *fasR* (+2.33 in log2) in *M. tuberculosis* induced only a weak effect on *fas* and *acpS* expressions (Rv2524c: +0.59; Rv2523c: -0.64) ?

We believe FasR is an exquisite modulator of the *fas-acpS* expression in *M. tuberculosis*, contributing to a tightly coordinated regulation of the FAS I and FAS II systems, that ultimately maintain lipid homeostasis in this microorganism. We already demonstrated that FasR activates the transcription of the *fas-acpS* genes (Mondino et al. *Mol Microbiol* 2013 89:372). In that work, we also proved that such expression activation is mediated via FasR-recognition of pseudo-palindromic sequences in the *fas-acpS* promoter region.

The transcriptomic analyses carried out by Rustad et al, compare the level of expression of the *fas-acpS* genes between a wild type *M. tuberculosis* strain and one that overexpresses FasR. However, the analysis of those experiments needs some careful considerations: 1) the threshold protein levels of FasR, above which no further induction can be detected, are not known, and 2) there is no experimental data about the actual levels of FasR protein in the wt vs the overexpressing strains. Therefore, it is hard to predict whether higher expression levels are to be expected, in the conditions of those assays. Furthermore, based on our earlier studies and on the regulatory mechanism that we now propose for FasR, we anticipate that the overexpression of this regulator should lead to an increased level of FAS I and therefore to higher levels of long-chain acyl-CoAs, which will consequently bind to FasR and inhibit its activator role. This feedback mechanism could well explain the results found by Rustad et al.

- The references cited in the first page of the introduction should be controlled, i.e. ref #5 before the #4 and the reference #3 do not appear to be the appropriate reference.

This has now been corrected.

- According the structure validation report for FasR/C20-CoA, the ligands (C20-CoA and C14) do not fit perfectly well in the electron densities. Could the authors show omit maps for ligands as supplementary figures, and also add in Table 1 for ligands some local density-based validation metrics, such as the real-space R-factor, the real-space correlation coefficient and/or the real-space Z-difference/observed. These additional data could be discussed in the text.

Thank you for the suggestions. We have now calculated and added OMIT maps for all ligands (new Supplementary Fig. 2) and real space correlation coefficients (comparing model-calculated maps vs final refined maps). We have included the RSCC figures just for the C₂₀-CoA and for both C₁₄ ligands in Table 1.

- Apparently, all constructions of the four mutants unfortunately contain the N-terminal part. It would have been great to solve the crystal structures of the “blocked” mutants (i.e. L106F and (L106F, V163F, L185F)), hoping to obtain an apo form and observing the effects on HTH organization and flexibility.

We agree with the reviewer. We have attempted these crystallizations, but with no success. The very mechanistic model is arguing against using crystallography as a means to obtain information on the true apo form: high flexibility (even “molten globule”-like traits) can easily explain difficulties to crystallize. NMR, in combination with calorimetry, could maybe be nice approaches to pursue, of course aiming for a different paper to be reported elsewhere.

- EMSA results with the mutant L98A are not very convincing and very similar to those with wild-type protein. Do you have an explanation?

There is some effect, although, we agree, less striking than the effect of Phe123. It could be because of the smaller volume of the side chain, hence a less drastic effect when it is down-sized to a single methyl group as in Ala (a Leu→Ala substitution reduces the volume by 78.1 Å³, whereas Phe→Ala implicates a considerably larger reduction of 101.3 Å³). In any case, we have now added a more extensive analysis by EMSA, using replicas and concentration ranges (new supplementary figures) which are summarised in new Table 2, showing that L98A also exhibits a similar trend as uncoupler, decreasing the C₂₀-CoA-triggered effect.

- Why only the mutant F123A has been tested by SPR? It would have been interesting to see if the mutant L98A also conserves the capacity to bind C20-CoA as the wild-type. In the same way, SPR should be used to further confirm by another experimental technique that the two “blocked” mutants have lost their capacity to bind to C20-CoA and therefore cannot be modulated in their DNA-binding activity (i.e. confirming the Figure 3b).

We concentrated on the SPR analyses for the F123A and the L98A mutants because they were both relevant to explain the molecular mechanism involved in the transmission of the signal from the effector to the HTH domain. Unfortunately, only F123A (and FasRwt) proved to be stable in the acidic buffer used to immobilize the protein to the SPR chip, which did not allow us to pursue this technique in a more comprehensive manner, as we would have wished.

After receiving your comments, we attempted analytical techniques (HPLC and/or mass spectrometry) to directly analyse the presence and identity of bound ligand(s) in the wt and in the FasR mutants. Technical hurdles precluded however reliable detection of the proteins and/or the effector ligand under the analytic conditions tested. Long-chain fatty acyl-CoA molecules exhibited poor behaviour in standard reversed-phase HPLC columns, not allowing for proper binding or elution. Considering this first obstacle, we then tried direct injection, both into an Orbitrap electron-spray ionization mass spectrometer, as well as on a MALDI-TOF/TOF instrument (attempting full-length native protein identification, and hence mass-shift when the acyl-CoA moiety binds). For reasons related to the very properties of FasR in terms of efficient ionization and flying behaviour within the MS tubes (dependent on the dimer’s mass, pI, native conditions to keep the folded

complexes in place, etc), we haven't been able to detect peaks reliably, precluding proper deconvolution procedures.

We do want to highlight at this point, that the much more extensive amount of data gathered from electrophoretic mobility shift assays, constitute a solid means to measure effector-triggered DNA-binding association modulation (especially comparing wt and mutant variants of FasR).

- I did not understand very well the interpretation of the mutant F123A. This mutant can accommodate C20-CoA while retaining its ability to bind DNA. I cannot imagine that this mutated protein can bind at the same time the C20-CoA effector molecule and the DNA. Surely, further experiments should be realized to understand this mutant (as for instance by SPR or X-ray).

Together with a much more comprehensive elaboration of the hydrophobic spine model (new data reported in Methods and in Supplementary Data 1; new Fig. 6 and Supplementary Fig. 10), we have now produced a more detailed explanation about the molecular bases of uncoupling mutants. As a means to summarise it schematically, we have added new Supplementary Fig. 11 to explain in a visual way.

Also, other mutants of this kind (allosteric effect uncouplers: they bind the effector ligand, but don't produce the allosteric effect in triggering DNA-dissociation) have been reported for other TetR-like regulators, so that we have now added further supporting examples (Muller et al 1995 Nat Struct Biol 2:693 which we have included in the References; and related work like Hecht et al 1993 J Bacteriol 175:1206, etc).

- Can the authors explicitly confirm in the text that no constraints were applied during the 10ns molecular dynamics simulation?

Yes, we have made this confirmation explicit now in the Methods section.

- I am not an expert in molecular dynamics simulation. However, it seems to me that the reliability of results should be assessed by comparing several MDS results, like "positive" and "negative" controls (for example, as positive control, an identical MDS with the C20-CoA and as negative control, may be a PEG400 molecule).

We think we understand what the reviewer is trying to point out here. Let us recall that the whole idea of using computational approaches to analyse a longer acyl-CoA ligand (longer than the one we crystallized), a ligand that is known to play biologically relevant roles, is because handling longer chains becomes increasingly difficult when it comes to performing *in vitro* biophysical studies (including X-ray crystallography, calorimetry, etc). The main question as we had stated it, is to see whether it would be plausible for the protein to handle longer chains that would literally traverse the entire tunnel and occupy the available space between the two FasR protomers in the dimer. If it were not, molecular dynamics would not converge toward a stable structure, all the way through the entire time lapse (in the jargon, the structure would have "exploded", literally unfolding apart because of the energetic cost in keeping a too large ligand bound within the site).

So, which would be a "positive" control? It would certainly not be C20-CoA, because we know that this is stable (its structure was actually solved experimentally). Therefore, a true positive control would be one that

would make “explode” FasR being difficult to define what that molecule would be, and as importantly, what additional information would that simulation be disclosing.

- To experimentally validate the MDS, SPR with the C26-CoA could be performed.

It should be noticed that working with long-chain acyl-CoAs is extremely challenging because these molecules form micelles above their critical micelle concentrations; CMC varies with the pH, the ionic strength and the length of the acyl moiety. Once micelles are formed, their presence has multiple consequences: 1) the concentration of free acyl-CoA changes, 2) the detergent properties of the micelle affect the protein integrity, 3) micelles also produce SPR signals (see *e.g.* Kamisaka et al 2011 *Biosci Biotechnol Biochem* 75:1135). Working with C20-CoA was already challenging because we couldn't go above 5 μ M, what makes really impossible the use of C26-CoA in this technique.

The computational work performed by using well established molecular dynamics simulations, allowed us to give a solid prediction about the probability of a very long-chain acyl-CoA to bind as a stable ligand to FasR.

- Please give the observed distance between HTH in the DNA-FasR complex structure.

The reviewer is right, we have now added this more clearly, directly in Fig 5 (before it was too hidden embedded within the mechanistic model [new Fig 7]).

- In discussion, the authors stated the volume delimited by the two protomers is unusually large in FasR. Compared to which tetR structures? Did they perform a survey of all solved TetR structures? Can we have some numerical values?

We have now added quantitative figures for a few TetR and TetR-like structures, especially selecting well-known models, so that side-to-side comparisons can be evaluated. To the best of our knowledge the number of TFR structures is currently over several hundreds, an exhaustive comparison seems unreasonable.

- In discussion, may be the following sentence should be rewritten “(in FasR-C14 complex) significantly higher flexibility of the HTH domains, as revealed by higher atomic displacement parameters and weaker electron density”. I think the HTH domains were well defined in the electron density whereas connecting parts between the HTH and the effector binding domains are more flexible and not well defined.

We thank the reviewer for this suggestion, the previous phrasing was just not detailed enough. We have now rephrased the sentence to make it more precise.

Calculating average isotropic B factors for each chain, and then plotting individual residues' B factors divided by the average (*i.e.* normalized atomic displacement parameters or nADPs), significantly higher figures are readily apparent for one of the HTH domains in FasR $_{\Delta 33}$ -C $_{14}$ (that of chain B) compared to the other, and also compared to FasR $_{\Delta 33}$ -C $_{20}$ -CoA's. This is visible in the region spanning HTH helices $\alpha 2$ and $\alpha 3$.

As you point out, higher flexibility of FasR $_{\Delta 33}$ -C $_{14}$ compared to FasR $_{\Delta 33}$ -C $_{20}$ -CoA was also revealed by larger nADPs in the $\alpha 6$ - $\alpha 7$ loops and the first half of helix $\alpha 7$, with weak electron density in one of the $\alpha 6$ - $\alpha 7$ loops (the one belonging to FasR $_{\Delta 33}$ -C $_{14}$ chain B).

- Deciphering how the allosteric signal in *tetR* family of regulators is transmitted from the effector binding domain to the HTH module remains highly challenging. To the best of my knowledge, nobody has yet demonstrated how this phenomenon happens. The authors suggest the existence of a hydrophobic spine common to TetR regulators and participating to the allosteric signal transduction upon effector binding. I am not really convinced by the transmission spine proposition. I do not think the results presented in the manuscript allow to extrapolate a general allosteric mechanism for all members of TetR family. Furthermore, the residues of the hydrophobic spine do not appear conserved across members of TetR family, according the supplementary Figure 1 and according images of Figures 6a, 6b 6c and the Supplementary Figure 5. It is mentioned that “a comprehensive MSA that confirms strict conservation is not shown”. This MSA should be provided to convince readers. There is clearly a lack of evidence precise and consistent enough to support the transmission spine hypothesis.

This is an interesting and important point here. First let us highlight that this is elaborated within the discussion section, as this is clearly part of the way we can interpret our (and others’) results. In this sense, we agree that we don’t want to be conclusive, a principle that holds for any interpretation in terms of models (be them mechanistic or otherwise). We have rephrased several subsections to be much clearer in this direction.

We are presenting sound experimental evidence to propose this interpretation, based upon the three crystal structures, structure-guided point-mutagenesis and functional analyses thereof. The multiple sequence alignment has now also been considerably extended (>2500 sequences) and included as Supplementary Data 1 file. We also believe, and argue within the ms, that our interpretation allows to better take into account the vast amount of evidence that the literature provides for TFR regulators.

Repeatedly described for TFRs the effector-triggered open/closed ‘pendular’ mechanism remained enigmatic in several respects. In particular, *i*- the closed-configuration structure has only been captured in structures of TFRs in complex with DNA, and not with the regulators alone in their effector-free state, challenging the paradigm of effector-triggered opening (Cuthbertson & Nodwell 2013 *Microbiol Mol Biol Rev* 77:440); *ii*- no simple shifts in individual residues appear to explain the mechanical bases of the alleged pendular movement; *iii*- a number of TFR mutants have been identified that either uncouple effector-binding from transcriptional induction (Muller et al 1995 *Nat Struct Biol* 2:693), or invert the effector’s action by triggering a tighter binding to DNA (Kamionka et al 2004 *Nucleic Acids Res* 32:842; Scholz et al 2004 *Mol Microbiol* 53:777), in both cases often implicating positions that are not directly involved in effector-binding.

The evidence to be interpreted includes ours as well as the previously outlined unresolved issues. The model we are now positing is consistent with all the reported behaviours of TFRs pointed out in the previous paragraph. Also explaining their capacity to bind more or less promiscuously to hydrophobic probes (Yu et al 2010 *J Mol Biol* 400:847) consistent with their tendency to complete a compact fold. Moreover, our structural insight nicely explains the behaviour of different synthetic pharmacophores, whereby the crucial ones in terms of inhibitory activity are actually interacting with hydrophobic spine residues in the *Mtb* drug target EthR (Willand et al 2009 *Nat Med* 15:537).

Other authors have offered coherent results and perspectives (*e.g.* Reichheld et al 2009 *PNAS* 106:22263;), albeit lacking a clear molecular basis that we now offer through structural analyses using mechanical strain scrutiny (instead of misleadingly seeking for positional shifts), which led us naturally to identifying the neighbouring residues able to exert that strain.

Directly addressing the reviewer’s concerns we have thus now :

- included all the references cited above within the revised ms, strengthening the validity of our interpretation model;
- performed a comprehensive multiple sequence alignment (MSA) including 2591 different TFR sequences, enriched in as diverse sequences as possible; we have now included the whole MSA as Supplementary Data 1 file;
- produced an amino acid conservation analysis which is more quantitative, based on the new MSA mentioned in the previous point, using a weighting algorithm to assign well-established hydrophobicity indices to each aligned position;
- clearly highlighted the hydrophobic spine residues, numerated according to the FasR sequence as a template (new panel a in Fig. 6), which can be easily and precisely extrapolated to other TFR proteins according to the MSA;
- improved the main and supplementary figures (new Fig. 6 and Supplementary Fig. 10) to show the spine residues in surface representation, better capturing their spatial continuity, and the role of effector ligands in completing the spine (also illustrated in new Supplementary Fig. 11).

Reviewer #3

General comments

This is a very nice paper investigating the function and mechanism of a key mycobacterial detail using a wide range of biochemical, biophysical and structural methods. Based on the results the authors propose a very plausible model for activation which will have wider implication on the field of bacterial repressors. Tuberculosis remains a major global challenge and hence basic research in academia will be vital to develop new and better therapies. The paper is therefore well suited for a more general journal such as Nature Communication.

Overall, the paper is well written, and of the highest technical level. The only drawback is the limited resolution of the FasR-DNA structure, however, the authors successfully avoided over-interpretation and given the well-known difficulties to obtain high-resolution crystals structures of (flexible) protein DNA complexes (which also makes these complexes almost impossible to tackle by cryoEM) this should not prevent publication. Overall, I recommend publication with minor revisions

However, there are a few points I would like the authors to consider for a revised submission

We appreciate the very positive feedback.

Specific comments

(0) The importance of structures in the search for new TB therapies could be mentioned in the introduction, there are several reviews for example from the Mtb structural genomics consortia that could be cited.

We have now highlighted and referenced the relevance of Structural Biology approaches, including the contribution of Structural Genomics consortia, boosting serious drug discovery efforts.

(i) Although the authors mention the mycobacterial EthR in one of their supplementary material, I would like to see more of a comparison. EthR should for example be in the sequence alignment as it would be interesting to know if the flexible N-terminus is present as well. In addition, both members of the TetR family seem to bind similar ligands, hence a more detailed comparison of the respective binding sites would be of interest. Also, least-squares superpositions of FasR with EthR could show if the the proposed mechanism could be employed by EthR (and other members of the family).

As per the reviewer's request, we have now added a more extensive comparative analysis with EthR.

We haven't added though EthR into the multiple sequence alignment of Supplementary Fig. 1a because that MSA was performed with TFRs that most resemble FasR in terms of 3D structure (adding also a few more sequences with unknown 3D structures, but also displaying highest sequence identities on their DNA-binding domains). EthR is quite different from FasR, as we now analyze in the revised Discussion. Both in terms of sequence identity (22%) as well as in structural superposition (~4.5 Å rmsd using the effector-binding domains).

It seemed thus preferable to build a separate pairwise alignment that we have now included in new Supplementary Fig 10, where the structural comparisons were also included.

(ii) The unbiased density of all ligands should be given in the supplementary material.

We have now added this in new Supplementary Fig. 2 (in response also to reviewer #2).

(iii) Although the FasR-DNA structure is of limited resolution, a least-squares superposition with known TetR-DNA structure (ribbon or trace) would be helpful. The resolution is not sufficient to determine if a ligand is bound or not.

As for the superposition, we have now added a new Supplementary Figure 9, following the reviewer's suggestion.

We however kindly disagree with the reviewer with respect to his/her second statement. We would like to highlight what we had included in this regard within the Methods section. Even at this resolution, difference Fourier is an extremely powerful (sensitive) tool: we have tested adding a ligand, and this produces clear evidence of atoms not being there (negative peaks of $>6\sigma$ using $[mF_{\text{obs}} - DF_{\text{calc}}]$ coefficients to calculate the difference map). Parts of the protein model of similar size and nearby position that were not initially included (such as the HTH domains, which were deliberately omitted from the MR search probe) result in $>6\sigma$ positive difference Fourier peaks, which would be the expected feature if the ligand were actually there. Low occupancy (e.g. $<20\%$) cannot be ruled out, but even if that were the case, it would be a quantitative significant difference from the DNA-unbound forms.

(iv) Overall the materials and methods are comprehensive, in particular in the crystallographic section, a few references may be missing and could be add, ie. T.-Y. Teng for cryocooling techniques, I. Ascone (2007) for

the beam line, Broennimann et al, 2006 for the Pilatus detector, and since Arcimboldo was used on a DNA binding protein, Proepper et al, (2014) could be added. If local non-crystallographic restraints were used, Uson I., et al. (1999) could be cited. Beam line (or in-house course) and wavelength used should be added to the crystallographic table in the main text.

The suggestions are much appreciated, and have now been incorporated into the revised version of the ms.

Reviewers' comments second round:

Reviewer #1 (Remarks to the Author):

The authors of the manuscript "Mycobacterium tuberculosis FasR senses very long fatty acyl-CoA with a tunnel and a transmission spine" have addressed several of the issues raised by myself and the other reviewers. I agree with the responses to my specific comments 1 and 2, and particularly appreciate the additional EMSA experiments that were performed enabling a quantitative analysis of the mutants. However, the following concerns still remain:

1. I made a general remark regarding the limited methodologies used in the work and the lack of novel biological insights in the context of the high standards and expected impact of papers typically published in Nature Communications. I respectfully don't completely agree with the response of the authors: "We believe that confirming for the first time, through structural studies, that very-long chain acyl-CoAs synthesized by the main fatty acid synthase complex, are the ligands that regulate de novo fatty acid biosynthesis in *M. tuberculosis* is a relevant finding with biological implications." Indeed, experimental evidence is presented that C20-acyl-CoA molecules are specific FasR ligands and affect the protein's function in vitro. However, for longer-chain acyl-CoAs, this is modeled but not experimentally shown. Furthermore and more importantly, no proof is presented that this ligand interaction results in the regulation of de novo fatty acid biosynthesis in vivo or that this has biological implications. The authors explain in their answers to the reviewers' comments that they have gathered additional data on the in vivo role of FasR for the bacterium's cell envelope's lipid composition and that a fasR mutant strain is less virulent in a mouse model, supporting the statement that this regulator has a biologically important role and that it is a viable target for drug development. It is disappointing that these data will not be available for the reader of this paper as they have been submitted to another journal. In my opinion, this would have made a convincing additional element to this manuscript.

2. The second general remark relates to the mechanistic model of the functioning of TetR family regulators. In response to my (and reviewer nr. 2) concerns regarding the novelty of this model in comparison to the models that have been proposed for other TetR-type regulators, the authors have extended the Discussion to clarify the claim that this work provides new insights causing a paradigm shift. First, it is stated in the response to the reviewers' comments: "The mechanistic model we are supporting is not the classical open-closure rearrangement of the DNA-binding domains, allosterically linked to association-dissociation (respectively) of the effector ligand. Instead, we are offering a better interpretation that integrates our own data and others': the apo structure is less compactly folded, whereas the effector-bound one is rigid (its fold being completed by the effector molecule itself)." Although the hypothesis of the apo structure being more flexible than the ligand-bound one is an interesting idea, there are no experimental evidences presented in this paper that confirm this hypothesis and provide more insights than previously published studies of ligand- and DNA-bound TetR regulators. In fact, the authors admit this in one of their responses to a comment of reviewer 2: "The very mechanistic model is arguing against using crystallography as a means to obtain information on the true apo form: high flexibility (even "molten globule"-like traits) can easily explain difficulties to crystallize. NMR, in combination with calorimetry, could maybe be nice approaches to pursue, of course aiming for a different paper to be reported elsewhere." As there is no proof for the existence of an apo form (see response to my specific comment nr. 3), I agree with reviewer nr 2 that it would have been a valuable element to structurally analyze some of the FasR mutants. NMR would indeed be a very appropriate technique to support the hypothesis as it enables to evaluate dynamics of protein structures. Second, the identification of the hydrophobic "spine" that is responsible for the transmission of an allosteric response upon ligand binding appears novel but I am not convinced by the data presented in the paper that this is conserved in all TetR-family regulators as claimed. It is written "A structure-curated multiple sequence alignment of >2500 TFR sequences ... allowed to pinpoint a subset of such hydrophobic residues as an extremely conserved cluster." (lines 360-362). But it is unclear which residues are conserved of the residues presented in Fig. 6a that make up the hydrophobic spine in FasR. For example, residues that correspond to Phe124, part of the spine, in close orthologs of FasR are not always hydrophobic (Suppl. Figure 1). The raw fasta data file presented in Suppl. Dataset 1, analyzing less closely related TetR-family regulators, is not easily interpretable for the reader and the explanation in the Methods does not provide insights

into how the authors reached the conclusion that this hydrophobic spine is conserved in all TetR regulators. In addition, the sequences of the structures presented in Fig. 6b, c and d and Supplementary Figure S12 do not even seem to be all present in this sequence alignment and it is unclear how the authors selected the residues shown in space-filling mode. These residues, and their conservation based on the structure-guided sequence alignment, should be presented as well.

In conclusion, two statements in the abstract of the paper are not supported by the data presented in the current version of the paper: "FasR...plays a central role in long-chain fatty acyl-CoA sensing and lipid biosynthesis regulation" and "The hydrophobic spine, conserved in all TetR-like transcription factors, offers new opportunities for drug discovery including anti-tuberculosis antibiotics." It is advised that these are toned down or that additional data are presented. Finally, as mentioned in my first review, this is a well-executed and high-quality work that merits publication. However, I have performed this review considering the rigorous standards imposed by Nature Communications. The above-explained concerns are not as prominent if the authors would consider a more specific journal, for example dedicated to protein structure-function studies, and toning down some of the strongest claims.

Reviewer #2 (Remarks to the Author):

Recommendation: Author Should Prepare A Minor Revision

General comment :

The authors have satisfactorily responded to my questions and the addition of new data (in particular EMSA and sequence analysis) greatly improves the quality of the manuscript and the strength of the message. However, I remain unconvinced about the hydrophobic transmission spine that can be generalized to all members of TetR family of regulators.

Considering that several different mechanisms for modulating the ability to bind to DNA by effector binding have been proposed in the TetR family members (including a pendulum-like motion of the DNA-binding domain as in TetR, QacR and CgmR, a coil-to-helix transition in helix alpha4 in HrtR, and a rigid body motion of the two monomers relative to each other in SimR), it is difficult to imagine a universal structural model for the transition between the repressing and induced conformations. In addition, even the mode of DNA-binding varies within the family as some members bind to DNA as pairs of dimers (e.g. QacR) whereas others bind as single dimers (e.g. TetR). Some members are not regulated by small-molecule effectors, but by proteins (e.g. DhaS, AmtR and SImA) or even by metal binding (e.g. SczA and ComR). And there are also many ways in which regulators of TetR family can interact with ligands (see Fig8 in Cuthbertson & Nodwell, 2013, Microbiol. Mol. Biol. Rev 77, 440). Taken all together, it is unlikely to have a common mechanism of modulation but rather that each protein is unique or, more probably, there are distinct subgroups for the conformational transition within the TetR family.

On the other hand, as a multiple sequence alignment of the 2591 sequences used in the manuscript has been provided, I could find sequences of the hydrophobic spine (based on the FasR sequence presented in Fig6a) and perform sequence conservation analysis (AL2CO and WEBLOGO). Thus, as expected, I observed the presence of a discontinuous sequence of 34-residue length, as showed in Fig6a, that is mainly composed of hydrophobic residues but also having several highly variable positions (i.e. positions 14, 15, 16, 17, 22, 24, 26, 28, 29, 32 and 34). The existence of the hydrophobic spine is certainly an interesting feature of sequences of the TetR family which appears to have never been highlighted before. But the link between this sequence feature and the conformational transition between the repressing and induced conformations has not actually been demonstrated so far. May be the hydrophobic spine comes from the hydrophobic core architecture conserved across members of TetR family (because sharing the same fold). Therefore, the hydrophobic transmission spine is just a hypothesis which should be presented as such throughout the manuscript.

Specific comments:

- The hypothesis of the hydrophobic transmission spine proposes that bound effector structurally complete the hydrophobic spine giving a more stable and compact fold whereas the apo form gives a sub-optimal fold with a hydrophobic spine broken. The DNA-binding form represents also a compact and stable fold with a optimal hydrophobic spine. However, the 3D images of hydrophobic spine (i.e. Fig6b,c,d and Supp Fig12a,b,c,d) show only liganded structures. As done in Supp Fig13, it would be interesting to compare hydrophobic spines in ligand-bound state vs apo-form (several apo forms available in the PDB) or vs DNA-bound form, in order to illustrate and verify the hypothesis.

- In the last paragraph of the discussion, please add more information on EthR in complex with piperidyl vs thienyl compounds, which PDBs? which compound names? Furthermore, as the binding of most EthR inhibitors were characterized (TSA and/or EPR), a comparison between piperidyl- vs thienyl-containing compounds would be done.

- The relationship between proteins of TetR family and eukaryotic protein kinases is not obvious considering the large divergence in regulatory dynamic mechanisms and in protein functions.

- Except for FasR (Fig.6a), sequences of hydrophobic spines were not shown. In figures illustration hydrophobic spines of other TetR members (i.e. Fig6 and Supp Fig.12), these sequences can be added.

- The weighted hydrophobicity index of each position of the hydrophobic spine should be given as supplementary data in addition to the sequence alignment.

- The hydrophobic scale of Kyte & Doolittle assigns a rather hydrophilic index to Trp and Tyr, -0.90 and -1.30, respectively, whereas these two residues are classified as hydrophobic in other hydrophobic scales. Taking into account the high presence of Trp and Tyr in the ligand binding cavity of some TetR family regulators (as in EthR), how would the hydrophobic spine be modified if another scale had been used?

- For the molecular dynamics simulation, I asked about some 'controls' not at the level of results but for the procedure used. For instance, to verify whether without a ligand the same MDS leads to a structurally unfolding, as expected. Or verify whether the use of another ligand effectively changes the dynamics.

=====

Reviewer #3 (Remarks to the Author):

Overall the authors have taken all (reasonable) suggestions by the reviewers into account. The manuscript has been much improved and I would support publication

Lara J et al. “*Mycobacterium tuberculosis* FasR senses very long fatty acyl-CoA with a tunnel and a transmission spine”

Point-by-point response to the reviewers’ comments.

Reviewer #1:

The authors of the manuscript “Mycobacterium tuberculosis FasR senses very long fatty acyl-CoA with a tunnel and a transmission spine” have addressed several of the issues raised by myself and the other reviewers. I agree with the responses to my specific comments 1 and 2, and particularly appreciate the additional EMSA experiments that were performed enabling a quantitative analysis of the mutants. However, the following concerns still remain:

1. I made a general remark regarding the limited methodologies used in the work and the lack of novel biological insights in the context of the high standards and expected impact of papers typically published in Nature Communications. I respectfully don’t completely agree with the response of the authors: “We believe that confirming for the first time, through structural studies, that very-long chain acyl-CoAs synthesized by the main fatty acid synthase complex, are the ligands that regulate de novo fatty acid biosynthesis in M. tuberculosis is a relevant finding with biological implications.” Indeed, experimental evidence is presented that C20-acyl-CoA molecules are specific FasR ligands and affect the protein’s function in vitro. However, for longer-chain acyl-CoAs, this is modeled but not experimentally shown. Furthermore and more importantly, no proof is presented that this ligand interaction results in the regulation of de novo fatty acid biosynthesis in vivo or that this has biological implications. The authors explain in their answers to the reviewers’ comments that they have gathered additional data on the in vivo role of FasR for the bacterium’s cell envelope’s lipid composition and that a fasR mutant strain is less virulent in a mouse model, supporting the statement that this regulator has a biologically important role and that it is a viable target for drug development. It is disappointing that these data will not be available for the reader of this paper as they have been submitted to another journal. In my opinion, this would have made a convincing additional element to this manuscript.

We are confirming for the first time using orthogonal experimental methods (EMSAs, SPR, crystallography, mutagenesis) that arachinoyl-CoA (a very long fatty acyl-bearing effector) regulates FasR function, as well as the details of its binding with molecular resolution.

Our study builds upon a biochemical and structural approach, and, very importantly, is directly linked to, and follows up on previous *in vivo* results that our group reported (Mondino et al Mol Microbiol 2013 89:372). This reviewer probably missed those data, which are however extremely relevant in proving the biological importance of FasR in Mycobacteria (thus mentioned as background knowledge in the Abstract and Introduction for the current study).

Furthermore, we are now providing the handling Editor with a manuscript where we report the key influence of FasR on *M. tuberculosis* cell wall lipid composition, as well as on the pathogen's virulence in the mouse model of infection. This work is ready to be submitted for publication.

2. The second general remark relates to the mechanistic model of the functioning of TetR family regulators. In response to my (and reviewer nr. 2) concerns regarding the novelty of this model in comparison to the models that have been proposed for other TetR-type regulators, the authors have extended the Discussion to clarify the claim that this work provides new insights causing a paradigm shift. First, it is stated in the response to the reviewers' comments: "The mechanistic model we are supporting is not the classical open-closure rearrangement of the DNA-binding domains, allosterically linked to association-dissociation (respectively) of the effector ligand. Instead, we are offering a better interpretation that integrates our own data and others': the apo structure is less compactly folded, whereas the effector-bound one is rigid (its fold being completed by the effector molecule itself)." Although the hypothesis of the apo structure being more flexible than the ligand-bound one is an interesting idea, there are no experimental evidences presented in this paper that confirm this hypothesis and provide more insights than previously published studies of ligand- and DNA-bound TetR regulators. In fact, the authors admit this in one of their responses to a comment of reviewer 2: "The very mechanistic model is arguing against using crystallography as a means to obtain information on the true apo form: high flexibility (even "molten globule"-like traits) can easily explain difficulties to crystallize. NMR, in combination with calorimetry, could maybe be nice approaches to pursue, of course aiming for a different paper to be reported elsewhere." As there is no proof for the existence of an apo form (see response to my specific comment nr. 3), I agree with reviewer nr 2 that it would have been a valuable element to structurally analyze some of the FasR mutants. NMR would indeed be a very appropriate technique

to support the hypothesis as it enables to evaluate dynamics of protein structures. Second, the identification of the hydrophobic “spine” that is responsible for the transmission of an allosteric response upon ligand binding appears novel but I am not convinced by the data presented in the paper that this is conserved in all TetR-family regulators as claimed. It is written “A structure-curated multiple sequence alignment of >2500 TFR sequences ... allowed to pinpoint a subset of such hydrophobic residues as an extremely conserved cluster.” (lines 360-362). But it is unclear which residues are conserved of the residues presented in Fig. 6a that make up the hydrophobic spine in FasR. For example, residues that correspond to Phe124, part of the spine, in close orthologs of FasR are not always hydrophobic (Suppl. Figure 1). The raw fasta data file presented in Suppl. Dataset 1, analyzing less closely related TetR-family regulators, is not easily interpretable for the reader and the explanation in the Methods does not provide insights into how the authors reached the conclusion that this hydrophobic spine is conserved in all TetR regulators. In addition, the sequences of the structures presented in Fig. 6b, c and d and Supplementary Figure S12 do not even seem to be all present in this sequence alignment and it is unclear how the authors selected the residues shown in space-filling mode. These residues, and their conservation based on the structure-guided sequence alignment, should be presented as well.

We realize that this reviewer, alas, is not considering the experimental (crystallographic) evidence provided by the FasR crystals bound to the shorter, C₁₄ moiety, a complex that exhibited marked asymmetry and a more pronounced protein flexibility compared to the one bound to C₂₀-CoA. And that shorter fatty acids such as C₁₄, are ineffective at triggering DNA-dissociation (Mondino et al Mol Microbiol 2013 89:372). Altogether, these are direct, experimental pieces of evidence supporting the proposed hypothesis of long-fatty acyl-dependent rigidification. This observation is further comforted by *in silico* modelling (using well established molecular dynamics simulations) as a means of informing anticipated stability of FasR in the presence of larger effector molecules, which are otherwise extremely difficult to manipulate in solution to surmounting solubility issues, as well as avoiding deleterious detergent effects on the protein.

It is then also pertinent to note that hypotheses must be built considering our own, as well as other groups' evidence: we are embedding our own observations in the context of a rather vast amount of data on numerous TFRs. The identification of a hydrophobic spine that physically connects both domains, and is completed by effector ligands in FasR and in many TFRs, is a direct observation we are making by analysis of the crystal structures we are reporting, as well as many other available structures of different TFRs.

As reviewer #2 stresses, the identification of such a hydrophobic spine is a novel observation, which offers insight to better understand experimental evidence gathered with many different TFR systems. In this same line of reasoning, and following up on experiments done with other TFRs such as TetR (now duly referred to in the main text), we are now adding further experimental evidence of higher protein flexibility in the apo vs C₂₀-CoA-bound form of FasR, probed by proteolysis sensitivity.

There are no structures of TFR regulators determined by NMR whatsoever and, more importantly, the reviewer does not suggest specific NMR experiments to be performed: how should we go about in evaluating the dynamics of FasR with and without ligands by NMR? To us this seems more like a technically very challenging aim, and not a feasible strategy for us to improve our current work, rather a line of research on its own. As for further crystal structures, we did respond to that in the first round of revision (as an answer to reviewer #2), and wish to insist here that we have attempted crystallizing mutants, but with no success.

Fig 6's caption states that panel (a) illustrates the "Hydrophobic spine amino acids in FasR. The colour code depicts residues of the DNA-binding domain in orange, the effector-binding domain in grey and the α 6- α 7 loop in cyan". In other words, this sequence corresponds to the spine constituents (hence, all these residues are conserved according to the 2591-sequences MSA). As a matter of fact, reviewer #2 understood this correctly, and actually went along and used it to do further sequence analyses that we appreciate (see further below).

Referring always to the way by which we identified the residues that make up the hydrophobic spine, reviewer #1 claims that "...the explanation in the Methods does not provide insights into how the authors reached the conclusion that this hydrophobic spine is conserved in all TetR regulators." However, the Methods section that describes what we did ("Structural bioinformatics to define the hydrophobic spine") is really detailed. We could modify it, if greater clarity were required, but reviewer #1 mistakenly states that we do not explain how we pinpointed the conserved hydrophobic positions, as we actually did.

Given the large number of TFR sequences and available structures, it is not of our intent to be comprehensive in illustrating the transmission spine for all cases. We believe it is useful to complement the multiple sequence alignment with further pdb structures in the detailed representation of the spine (Figs 6b-d and Supplementary 13 [previous S12], combined with Supplementary Data sets 1 and 2). Residues shown in molecular surface (not space-filling) in Figs 6b-d and Supp 13 correspond to all the spine residues, as defined in the Methods section and depicted in Fig 6a. Further critiques about the identification and analysis of the hydrophobic spine, are duly addressed in response to the suggestions that reviewer #2 formulated (see below).

Reviewer #2:

General remarks

The authors have satisfactorily responded to my questions and the addition of new data (in particular EMSA and sequence analysis) greatly improves the quality of the manuscript and the strength of the message. However, I remain unconvinced about the hydrophobic transmission spine that can be generalized to all members of TetR family of regulators.

Considering that several different mechanisms for modulating the ability to bind to DNA by effector binding have been proposed in the TetR family members (including a pendulum-like motion of the DNA-binding domain as in TetR, QacR and CgmR, a coil-to-helix transition in helix alpha4 in HrtR, and a rigid body motion of the two monomers relative to each other in SimR), it is difficult to imagine a universal structural model for the transition between the repressing and induced conformations. In addition, even the mode of DNA-binding varies within the family as some members bind to DNA as pairs of dimers (e.g. QacR) whereas others bind as single dimers (e.g. TetR). Some members are not regulated by small-molecule effectors, but by proteins (e.g. DhaS, AmtR and SlmA) or even by metal binding (e.g. SczA and ComR). And there are also many ways in which regulators of TetR family can interact with ligands (see Fig8 in Cuthbertson & Nodwell, 2013, Microbiol. Mol. Biol. Rev 77, 440). Taken all together, it is unlikely to have a common mechanism of modulation but rather that each protein is unique or, more probably, there are distinct subgroups for the conformational transition within the TetR family.

On the other hand, as a multiple sequence alignment of the 2591 sequences used in the manuscript has been provided, I could find sequences of the hydrophobic spine (based on the FasR sequence presented in

Fig6a) and perform sequence conservation analysis (AL2CO and WEBLOGO). Thus, as expected, I observed the presence of a discontinuous sequence of 34-residue length, as showed in Fig6a, that is mainly composed of hydrophobic residues but also having several highly variable positions (i.e. positions 14, 15, 16, 17, 22, 24, 26, 28, 29, 32 and 34). The existence of the hydrophobic spine is certainly an interesting feature of sequences of the TetR family which appears to have never been highlighted before. But the link between this sequence feature and the conformational transition between the repressing and induced conformations has not actually been demonstrated so far. Maybe the hydrophobic spine comes from the hydrophobic core architecture conserved across members of TetR family (because sharing the same fold). Therefore, the hydrophobic transmission spine is just a hypothesis which should be presented as such throughout the manuscript.

Starting from the very last sentence of this reviewer's general remarks, which we appreciate, we have rephrased several sections of the manuscript to make it clearer that this interpretation is a working hypothesis, albeit one that explains all reported data. Integrating the reviewer's suggestions into the writing,

- i- we have modified one phrase on the Abstract, better defining the transmission spine. Namely, highlighting observed facts such as that it does correspond to the hydrophobic core that maintains the protein fold, which we literally see in our crystal structures, as well as in structures of each and every other TFR proteins that we analyse. Such a hydrophobic core connects both domains and the binding of the effector ligand is observed to complete this core into a continuous spine in FasR and all TFRs analysed.
- ii- the whole analysis of the transmission spine is now described in much greater detail, as requested, both in the Discussion and the Methods sections. Namely by:
 - * explicitly stating that this spine corresponds to the protein-folding hydrophobic core, observed to connect both domains and to be completed by effector-binding;
 - * further clarifying the methods to define it in detail;
 - * reporting the spine sequences for all the TFRs analysed;
 - * refining the multiple sequence alignment to correct for minor errors in the structural superpositions (this also responds to one critique from reviewer #1 concerning residue 124, which is indeed always hydrophobic);
 - * keeping 32 out of the 35 amino acids that constitute the defined spine so as to exclusively maintain all those that conform a physically continuous cluster; and,
 - * double checking this definition by applying four different hydrophobicity index scales (new Supplementary Data 2).

Concerning the reviewer's analysis of sequence conservation, we appreciate the attached sequence logo and the questions regarding more and less variable positions. In coincidence with the reviewer's first conclusion, we are glad to confirm that all the spine positions are indeed occupied by hydrophobic residues, either absolutely or at the very least heavily biased towards this type of residue. More specifically, the eleven positions that the reviewer remarks as exhibiting greater variability, are all occupied by hydrophobic residues in a decidedly vast majority of cases, as neatly revealed in her/his attached logo. We thus believe our main point is not actually questioned.

In any case, and thanks to the reviewer's point, a slight refinement in the precise choosing of residues has been made, considering that positions 28, 29 and 31 bear residues that do not conform a physically continuous spine together with all the other selected amino acids. We have now stated this in the main

text and methods, excluding them from the defined spine (positions 28 and 29 were among the 11 positions that the reviewer pinpoints as more variable). By looking at the other 9 positions, the structures do suggest functional reasons as to why they might be more variable. Positions such as 14, 15, 16 and 17: in this central group, actually the first and the last are more variable than other positions of the motif, but always hydrophobic. Only in positions 15 and 16 His and Gln respectively, appear with much lower yet detectable frequencies. In FasR these two positions correspond to Val95 and Leu98, within helix $\alpha 4$, approximately in the middle of this long helix. Both sit right next to (at VDW contact distance from) the volume where all ligand binding sites are shared among TFRs. In some TFRs positions 15 and/or 16 evolved to interact with polar groups of specific effector ligands. For example, (i) FadR (PDB 6el2) : at position 15, Asn69 H-bonds to Arg73 which itself binds the ribose O3' phosphate (in FadRs the whole nucleotidic portion of CoA is well bound and defined in the crystal structures [see Yeo et al NAR 2017]); (ii) TetR (PDB 2xpw) : at position 16, His64 H-bonds directly to the oxy-tetracycline effector molecule (in this case quite polar in nature) N ϵ -O3 and N ϵ -O21; (iii) RamR (PDB 3vvy): at position 15 Lys63 interacts via its side chain N atom with atom N23 on the bound ethidium effector molecule (which is hydrophobic but exhibits a few key polar groups). Position 22 shows infrequent Ser, Thr and His residues, it corresponds to FasR Ile127, and as mentioned before, this position sits at the end of helix $\alpha 5$, just facing, and exactly at the same level, as position 16 on $\alpha 4$ (at VDW distance from the shared volume of all ligand binding sites). Positions 24 and 26, which in FasR correspond respectively to Val152 and Leu166, are placed at the N- and C-tips of helix $\alpha 7$. Infrequent polar residues at these positions once again correlate with singular interactions with polar groups in bound molecules (as in PDB 2hyt, where Q124 at position 24 is in contact with the fortuitously bound ligand EDO; or yet position 26 His139 in PDB 2qib, contacting the fortuitously bound ligand PEG). This scenario seems to hold for all these more variable positions, as adaptations to peculiarities in the diverse array of effector ligands that bind in the pocket, even though the heavy bias towards hydrophobic residues is by and large maintained in thousands of TFR sequences.

As for the general remark regarding a common regulatory mechanism for TFRs. We believe it is very important to highlight that the entire TFR family is homologous from an evolutionary perspective. A common functional regulation mechanism seems thus the most plausible scenario. Even though specific variations did arise, evolving alongside the wide diversity of effectors that the family is able to sense. All TFRs, despite sequence variations, share an astonishingly similar structural fold, a sequence-homologous DNA-binding domain and often a clearly homologous effector-binding domain as well. Even in cases where the latter identity becomes undetectable, homology is always obvious at the structural level. With such a similar two-domain structural architecture and strictly identical topology, does it seem sensible for a large variety of fundamentally distinct mechanisms to have evolved? Or, as this reviewer puts it, yet that each TFR protein would be unique in terms of mechanistic workings? We do not think so, and would greatly appreciate receiving this reviewer's comments with this evolutionary context in mind.

Contemplating the examples that this reviewer cites to challenge a common mechanism scenario, we actually observe that they are rather well consistent with the hydrophobic spine hypothesis:

- a- Concerning the proposed "pendular movement" of HTH domains, put forward for QacR and for many other TFR regulators, it is pertinent to insist on the concepts we elaborate within the ms, and that we have highlighted before in response to the reviewer's comments:
 - the effector-triggered open/closed 'pendular' mechanism has been put forward many times, but remains enigmatic in several respects;
 - the closed-configuration structure has only been captured in structures of TFRs in complex with DNA, and not with the regulators alone in their effector-free state, challenging the

paradigm of effector-triggered opening (Cuthbertson & Nodwell 2013 *Microbiol Mol Biol Rev* 77:440); in fact, structures claimed to be apo, are either open (i.e. similar to effector-bound ones) or, if closed, exhibit their HTH-domains trapped in position by crystal packing contacts;

- no simple shifts in individual residues appear to explain the mechanical bases of the alleged pendular movement;
- a number of TFR mutants have been identified that either uncouple effector-binding from transcriptional induction (Muller et al 1995 *Nat Struct Biol* 2:693), or invert the effector's action by triggering a tighter binding to DNA (Kamionka et al 2004 *Nucleic Acids Res* 32:842; Scholz et al 2004 *Mol Microbiol* 53:777), in both cases often implicating positions that are not directly involved in effector-binding.

All in all, it seems as thus the closed configuration of the pendulum is not actually a stable, discrete state, and that a number of residues that don't bind the effector moiety do participate in allosteric transmission. All this is well explained by the hydrophobic spine.

- b- In the case of HrtR (Sawai et al 2012 287:30755), the stabilization of $\alpha 4$ as a continuous helix implicates several conserved hydrophobic spine residues binding to the incoming heme porphyrin group (see Fig. 1 below). This is actually a nice and further example of the validity of our hypothesis. On top of the conserved mechanism, residues His72 and His149 (respectively on HrtR helices $\alpha 4$ and $\alpha 8$) are specific variations of this regulator evolved to coordinate the iron atom on the heme moiety, and also increasing the stabilization of the open, rigid conformation, probably explaining why the absence of ligand promotes such an extreme destabilization of helix $\alpha 4$. On the other hand the apo HrtR structure is almost identical to the DNA-bound form, but crystal packing is very likely trapping the closed HTH conformation given the strong contacts engaging the HTH domains established among the two dimers per asymmetric unit (head-to-tail HTH:effector-binding domain interactions) plus the several HTH:HTH interactions with neighbor molecules in adjacent unit cells of this orthorhombic crystal.

Figure 1. Almost all hydrophobic spine residues are conserved in HrtR, depicted in molecular surface representation here. Note that they connect both domains as in FasR and all other TFRs analysed, and leaving a cavity where the heme effector binds (shown in sticks coloured by atom type). The colour-code of the spine is the same as the one used in our ms Fig. 6. The only residues that don't align well are those sitting on FasR helix $\alpha 7$, given that this helix is quite shorter in HrtR with most of its C-terminal portion present as an unstructured loop, adapted to the large heme ring.

- c- That some TFR members are regulated by other proteins, such as SlmA, stabilized in the closed configuration via binding to FtsZ (Schumacher *PNAS* 2016 113:4988), is not contradictory with

the hydrophobic spine transmission mechanism. Moreover, in that work Schumacher & Zeng state: “In some of the structures [of SlmA] the DNA-binding domains are significantly disordered, but in others a conformation very similar to the DNA-bound conformation was trapped in the crystal. It is thought this flexibility in the apo form, as in other TetR proteins, allows SlmA to adjust optimally and dock on the DNA (43). Therefore, apo SlmA does not adopt a specific structural state; instead, the DNA-binding domains are dynamic when SlmA is not complexed to specific DNA (Fig. 2B). DNA binding locks in a specific conformation, as revealed by the finding that all SlmA structures have an essentially identical conformation in multiple SlmA–DNA structures (34).”

If we add to this that 1- the DNA-binding-competent, closed conformation of SlmA engages most of the hydrophobic spine residues as per our definition, observed to be obliterating any internal cavity within SlmA’s effector-binding domain; and that 2- to bind FtsZ, SlmA must previously bind to DNA locking it in the closed conformation; we can see this as a nice confirmatory example for the common hydrophobic spine model.

Specific comments

- The hypothesis of the hydrophobic transmission spine proposes that bound effector structurally complete the hydrophobic spine giving a more stable and compact fold whereas the apo form gives a sub-optimal fold with a hydrophobic spine broken. The DNA-binding form represents also a compact and stable fold with a optimal hydrophobic spine. However, the 3D images of hydrophobic spine (i.e. Fig6b,c,d and Supp Fig12a,b,c,d) show only liganded structures. As done in Supp Fig13, it would be interesting to compare hydrophobic spines in ligand-bound state vs apo-form (several apo forms available in the PDB) or vs DNA-bound form, in order to illustrate and verify the hypothesis.

As we state within the main text of the manuscript,

Binding the effector stabilizes an open configuration of FasR, which does not necessarily imply that the effector mechanically triggers a closed to open transition. If the latter mechanism were true, the ligand-free structure should exhibit a closed, DNA-binding competent configuration. A ligand-free form of FasR could not be crystallized, but turning our attention to the many other available TFR structures, those that exhibit no ligand almost exclusively correspond to the open form^{9,11} contradicting the predicted outcome. The very few apo structures with a closed configuration have unexplained density within the binding pocket (pdb 2FX0) and/or reveal crystal packings that fortuitously fix HTH domains strongly in place (e.g. the hexagonal form of 2FX0 and the centred monoclinic 1T33). To the best of our knowledge, the closed configuration has only been captured reliably in crystal structures of TFRs in complex with DNA. Additional evidence further challenge a simple open/closure mechanism: i- no simple positional shifts of individual residues can explain the mechanical bases of the alleged pendular movement; ii- a number of TFR mutants have been identified that either uncouple effector-binding from transcriptional induction³¹, or invert the effector’s action by triggering a tighter binding to DNA^{32,33}, in both cases often implicating positions that are not directly involved in effector-binding.

it is thus not feasible at this point to use “apo” structures of other TFRs to simply compare structures side-by-side and draw conclusions, which we think might be misleading : such alleged “apo” structures trapped by crystallography typically correspond to the HTH-open form, similar to effector-bound states; and the very few closed states (as underlined in the paragraph above) have serious features that cast doubt to producing candid analyses.

- In the last paragraph of the discussion, please add more information on EthR in complex with piperidyl vs thienyl compounds, which PDBs? which compound names? Furthermore, as the binding of most EthR

inhibitors were characterized (TSA and/or EPR), a comparison between piperidyl- vs thienyl-containing compounds would be done.

We thank the reviewer for raising this point. The phrasing we had used misled the reader to the idea that thienyl- and piperidinyl-containing compounds were separate. In the work analysed (Willand et al 2009 Nat Med 15, 537) the authors come up with EthR inhibitor molecules containing both pharmacophores simultaneously. We hope the new rephrasing elaborate these features clearly now, specific PDBs analysed are now made explicit.

- The relationship between proteins of TetR family and eukaryotic protein kinases is not obvious considering the large divergence in regulatory dynamic mechanisms and in protein functions.

We have now rephrased this sentence to highlight the fact that these two groups of proteins (ePKs and TFRs) are not homologous, indeed. However, we do want to keep this observation, because even if only analogous, it seems to us as a very important example of convergent evolution: the hydrophobic transmission spines in ePKs are well documented, and are implied in activity regulation of these bi-lobed enzymes, including via substrate-mediated completion. The fact that such structural insight has been used to develop very successful inactive-stabilizing compounds that maintain the regulatory hydrophobic spine in a broken state (Gleevec has been a game-changer to treat chronic myelogenous leukaemia, by targeting the BCR-Abl tyrosine-kinase), seems to us as an extremely exciting avenue to explore in FasR (and possibly TFRs in general, including EthR).

- Except for FasR (Fig.6a), sequences of hydrophobic spines were not shown. In figures illustration hydrophobic spines of other TetR members (i.e. Fig6 and Supp Fig.12), these sequences can be added.

All the sequences have now been added in both figures.

- The weighted hydrophobicity index of each position of the hydrophobic spine should be given as supplementary data in addition to the sequence alignment.

This has now been included in Supplementary Data 2 (Excel table).

- The hydrophobic scale of Kyte & Doolittle assigns a rather hydrophilic index to Trp and Tyr, -0.90 and -1.30, respectively, whereas these two residues are classified as hydrophobic in other hydrophobic scales. Taking into account the high presence of Trp and Tyr in the ligand binding cavity of some TetR family regulators (as in EthR), how would the hydrophobic spine be modified if another scale had been used?

Three different hydrophobicity index scales have now been compared, which rely on different criteria. Results are almost identical. We have included this in Supplementary Data 2 as well.

- For the molecular dynamics simulation, I asked about some 'controls' not at the level of results but for the procedure used. For instance, to verify whether without a ligand the same MDS leads to a structurally unfolding, as expected. Or verify whether the use of another ligand effectively changes the dynamics.

We have now simulated similar trajectories in the absence of effector bound, and show the results in new Supplementary Figure 9 (extending main Fig. 4). The results nicely confirm our transmission spine hypothesis, with concomitant lesser stability for the effector-binding domain (as revealed by a significantly larger intra-domain rmsd for apo FasR) and larger wiggling of the DNA-binding domains (as revealed by the substantially larger rmsd of the moving DBDs when the effector-binding domains are superimposed, comparing the apo vs the C₂₆-CoA-bound form; whilst intra-DBD rmsd is still low).

Reviewer #3

Overall the authors have taken all (reasonable) suggestions by the reviewers into account. The manuscript has been much improved and I would support publication

We appreciate the positive feedback.

REVIEWER COMMENTS third round

Reviewer #2 (Remarks to the Author):

I greatly appreciate the effort made by the authors to address my comments in detail and to bolster the manuscript with new data, in particular, the trypsin proteolysis experiments and further computational analysis (i.e MD simulations and sequence/structure bioinformatics). A continuous hydrophobic spine spanning effector-binding domain through DNA-binding domain is now more convincingly presented as a conserved structural feature of the TetR family. The role played by this hydrophobic spine, therefore denoted as the transmission spine, in the allosteric modulations of a TetR-family protein, for controlling its DNA-binding ability, is certainly an interesting insight. However, taking into account the diversity of TetR-family proteins, their abundance, their evolution and the fact that many members have not yet been well characterized, I think the authors should approach the generalization of the transmission spine concept to the whole TetR family with caution. This generalization would presumably require further experimental evidences. In addition, our current understanding of allosteric movements in TetR family is largely based on solved crystal structures. The numerous static structures available have limitations and fail considering all the different states (i.e. apo, liganded or bind to DNA). Furthermore, there are a huge number of TetR-family members (over 200,000 sequences identified in public databases), it is likely that exceptions to this concept may exist. In the new version of the manuscript, the transmission spine concept is now mostly presented and discussed as a working hypothesis, thus I recommend the manuscript for publication. Can I just kindly ask the authors to consider my last three comments?

- My first comment concerns the "homology" of the entire TetR family (then answering to authors' comment formulated in their reviewing responses). Homology is clear and undoubtedly, when common structural fold is associated with high level of sequence identity, especially between proteins/genes of two different species. Otherwise, when proteins share a common structure, but without significant sequence identity (as for a lot of TetR-family members), this can result of divergence from a common ancestor (remote homology) or of convergence to a favorable fold (no common ancestor and, *stricto sensu*, no homology).

There are 52 TetR proteins in *Mycobacterium tuberculosis* according Balhana et al. (2015, BMC Genomics). Can we claim with certainty that the 52 Mtb TetR proteins descend all from ONE common ancestor? Can we claim with certainty that Mtb FasR is homologous, for instance, to Mtb EthR and Mtb KstR? In my personal point of view, no, because we cannot completely exclude the possibility of a convergent evolution or the existence of a few common ancestors. In general, the term "homology" should be used sparingly, only when the existence of a common ancestor is undeniable. Otherwise, the term "similarity" should be preferred instead. However, it is commonly accepted (and I agree with) that TetR family regulators share DISTANT homology considering their common fold and the sequence similarity exhibited only by their HTH motifs. In contrast, their ligand-binding domains are strong divergent (see the domain organization of the PFAM family TetR_N, PF00440). Finally, even if we assume the homology of the whole TetR family, very likely via gene duplications (Voordeckers et al., 2015, Curr Opin Biotech), the absence of sequence similarity outside the HTH motifs suggests the proteins certainly had enough evolutionary time to acquire novel functions or even to modify their allosteric regulatory mechanism.

- I agree with the authors that an effector/ligand triggering closed to open transition is in opposition with most of the available data. Furthermore, in agreement with the allosteric mechanism presented for FasR, the flexibility of the apo form for many different TetR-family proteins is well documented. By experience, I know also the difficulty to produce crystals of unliganded (apo) TetR-family proteins. A general concept proposes that the apo form is a multi-state form, due to the inherent flexibility in the connection between ligand-binding and DNA-binding domains, whereas the binding with DNA or with effector/ligand traps the protein in a closed or open state, respectively (described for example in Yu et al, 2010, JMB, but also in several other papers). Accordingly, and theoretically, an apo form could be solved in various states, from closed to open state. "Theoretically", because crystallization involves stabilization and reduced flexibility, which is difficult for a protein expected unstable and flexible. Therefore, several crystal structures of TetR-family regulators declared to be in apo form can indeed contain extra electron densities in the ligand-binding cavity, suggesting the presence of unidentified ligands, and

consequently these crystal structures are mostly in open dimeric state. Likewise, crystal packings can explain some other apo forms. Nowadays, there are more than 250 structures of a TetR-family protein, with many structures declared in apo form. Did the authors performed an exhaustive survey of the data? From memory, I think PDB id 4JKZ (*M. smegmatis* Ms6564 in apo form) is in a closed state. Lines 336 to 345 on page 13 should be carefully controlled.

- Five years ago, I analyzed ligand-binding cavities of the TetR-family structures with the aim to compare them (unpublished data). Immediately I was confronted with the difficulty of such an analysis due to the great diversity in cavities but also in the ligand binding sites. Indeed, all the ligands do not bind in the cavity. Thus, I think we have to be careful in claiming a common allosteric mechanism for the TetR family of regulators. I am not saying it is not possible. I am just asking for more experimental evidences. The heterogeneity of TetR-family proteins should prompt us to prudence for a common allosteric mechanism (Colclough et al. 2019, BMC Genomics).

René Wintjens
Free University of Brussels

Lara J et al. “Mycobacterium tuberculosis FasR senses long fatty acyl-CoA through a tunnel and a hydrophobic transmission spine”

Point-by-point response to the reviewers' comments.

Reviewer #2:

I greatly appreciate the effort made by the authors to address my comments in detail and to bolster the manuscript with new data, in particularly, the trypsin proteolysis experiments and further computational analysis (i.e MD simulations and sequence/structure bioinformatics). A continuous hydrophobic spine spanning effector-binding domain through DNA-binding domain is now more convincingly presented as a conserved structural feature of the TetR family. The role played by this hydrophobic spine, therefore denoted as the transmission spine, in the allosteric modulations of a TetR-family protein, for controlling its DNA-binding ability, is certainly an interesting insight. However, taking into account the diversity of TetR-family proteins, their abundance, their evolution and the fact that many members have not yet been well characterized, I think the authors should approach the generalization of the transmission spine concept to the whole TetR family with caution. This generalization would presumably require further experimental evidences. In addition, our current understanding of allosteric movements in TetR family is largely based on solved crystal structures. The numerous static structures available have limitations and fail considering all the different states (i.e. apo, liganded or bind to DNA). Furthermore, there are a huge number of TetR-family members (over 200,000 sequences identified in public databases), it is likely that exceptions to this concept may exist. In the new version of the manuscript, the transmission spine concept is now mostly presented and discussed as a working hypothesis, thus I recommend the manuscript for publication. Can I just kindly ask the authors to consider my last three comments?

We thank the reviewer for his positive feedback, and for all the constructive suggestions and critiques along the reviewing process.

We do understand that, especially when dealing with such huge families, making generalizations is not an easy task, and probably never quite possible. We must progress by induction, always observing facts from individual (or subsets of) protein members, and then at the most attempting to grasp general mechanisms.

We thus agree with the reviewer in that caution is needed, avoiding unnecessary extrapolations that would actually require further evidence. We are also glad that the reviewer states that we are now presenting and discussing the transmission spine concept as a working hypothesis, and not as an assertion.

Specific responses below, for each of the comments:

- My first comment concerns the "homology" of the entire TetR family (then answering to authors' comment formulated in their reviewing responses). Homology is clear and undoubtedly, when common structural fold is associated with high level of sequence identity, especially between proteins/genes of two different species. Otherwise, when proteins share a common structure, but without significant sequence identity (as for a lot of TetR-family members), this can result of divergence from a common ancestor (remote homology) or of convergence to a favorable fold (no common ancestor and, stricto sensu, no homology).

There are 52 TetR proteins in Mycobacterium tuberculosis according Balhana et al. (2015, BMC Genomics). Can we claim with certainty that the 52 Mtb TetR proteins descend all from ONE common ancestor? Can we claim with certainty that Mtb FasR is homologous, for instance, to Mtb EthR and Mtb KstR? In my personal point of view, no, because we cannot completely exclude the possibility of a convergent evolution or the existence of a few common ancestors. In general, the term "homology" should be used sparingly, only when the existence of a common ancestor is undeniable. Otherwise, the term "similarity" should be preferred instead. However, it is commonly accepted (and I agree with) that TetR family regulators share DISTANT homology considering their common fold and the sequence similarity exhibited only by their

HTH motifs. In contrast, their ligand-binding domains are strong divergent (see the domain organization of the PFAM family TetR_N, PF00440). Finally, even if we assume the homology of the whole TetR family, very likely via gene duplications (Voordeckers et al., 2015, Curr Opi Biotech), the absence of sequence similarity outside the HTH motifs suggests the proteins certainly had enough evolutionary time to acquire novel functions or even to modify their allosteric regulatory mechanism.

We understand the point that the reviewer makes, and agree to follow his advice of greater caution.

More specifically we have further toned down the last sentence in the Abstract, thus avoiding the misleading sense of universal validity of the regulatory mechanism observed in *Mtb* FasR. We also added an explicit statement in the Discussion, laying down a clear scenario of different mechanisms still possible and necessitating further experimental validations.

As a side comment, only for the sake of this rewarding discussion with the reviewer: we believe it is a widely accepted notion that homology does not require high level of sequence identity. Any level of detectable identity is a proof of evolutionary relatedness (hence, of homology). This is why there is no such thing as high or low homology, two proteins are either homologous or they are not (at difference with levels of identity or similarity, which instead can be higher or lower according to several metrics). Homology of course can take the form of orthology, as the reviewer remarks, when it comes to identifying homologous genes/proteins in two different species. Paralogy is nevertheless a widely accepted form of homology as well, whereby a gene is duplicated (one duplication event, or several subsequent duplications), such that the two or more related genes then diverge and become paralogs. In any case homology always implies common ancestry.

Having said that, as levels of sequence identity become lower and lower, due to higher divergence (either because of faster mutation rates, stronger selective pressure on a rapidly changing environment, or yet just because of a larger amount of time passed by since radiating from the most recent common ancestor), a twilight zone is indeed encountered, where it becomes more and more difficult to distinguish true homology (hence, high divergence) from analogy (i.e. evolutionary convergence: achieving a similar 3D fold in proteins, but starting from independent, unrelated ancestors).

Coming back to TFRs, the N-terminal HTH domains are undoubtedly homologous, as a consequence of what, all TFR DNA-binding domains fall into a single PFAM family as the reviewer remarks. On the other hand, the C-terminal effector-binding domains can indeed exhibit undetectable sequence identity. This fact leads to a rather large number of different PFAM families that correspond to such diverse domains (named TetR_C followed by a number). The reviewer is thus right in that we cannot be absolutely conclusive about their single ancestry.

Not claiming conclusiveness, we do wish to note that structural similarity, readily quantitated by 3D superposition algorithms, has been shown to be a valid means of identifying homology even when sequence identity is no longer recognizable (Orengo & Thornton 2005 Annu Rev Biochem 74:867-900). When such structural relatedness is reinforced with identical topology (sequential disposition of secondary structure elements) and similar function (such as, in the case of TFRs, binding signal effector molecules, no matter how diverse they are), it is taken as a very strong signature of common ancestry.

We don't wish to delve in this direction within this manuscript though, as it deserves a proper phylogenetic analysis on its own, well beyond the scope of this current work. Let us briefly add here, that besides the structural homology evidence, sequence-based homology appears indeed to be still recognizable among all TetR_C domains in PFAM :

- i- Genes predicted to encode a TFR HTH domain (PFAM family TetR_N PF00440) currently represent >155,000 sequences, approximately half of which exhibit similar architectures comprising N-term TetR_N and a C-terminally located TetR_C domain. The other half are clustered into a single architecture comprising no defined C-terminal domain, albeit all similar in length to a typical, full-sized TFR. Towards the C-terminal portion, this architecture only identifies two "low complexity" segments. However, the representative of this large group of sequences is Z9JXQ3_9MICO from *Brachy bacterium phenoliresistens*, and performing a manual analysis (PSI-Blast) readily reveals that in Z9JXQ3_9MICO, the TetR_N domain is actually fused to a TetR_C29

- domain (several other sequences were manually picked out from this architecture group, and they all end up aligning to one or the other among the TetR_C domains at their C-terminal portions).
- ii- Practically all TFR effector-binding domains are clustered into a single PFAM clan (named TetR_C CL0174, currently comprising >74000 domains; this number will likely evolve including many more, when the more distantly related ones such as Z9JXQ3_9MICO, are correctly classed into TetR_C* families). The very definition of clans in PFAM states that “a clan contains two or more PFAM families that have arisen from a single evolutionary origin” (Finn et al doi:10.1093/nar/gkj149). In other words, all the hidden Markov model profiles that represent each of the TetR_C* families’ multiple sequence alignments, were pairwise-compared (using the Profile Comparer [PRC] approach, Madera M doi:10.1093/bioinformatics/btn504) resulting in detectable relatedness. Only two TetR_C families are not included among the 36 that currently compose the TetR_C CL0174 clan, and those are TetR_C12 (PF16914, ~80 sequences) and TetR (PF13972, ~1000 sequences). Having said that, these latter two families are indeed homologous to some of the ones included in the CL0174 clan (I have rapidly done it using PRC), it is likely that next updates of the PFAM database will eventually reflect this.
 - iii- Even if we consider that such sequence similarities among effector-binding domains (detectable using HMM profiling) are too low and rather represent examples of convergence, it then seems quite improbable that such a large number of different, non-homologous effector-binding domains, would have fused repeatedly, through independent fusion events, to a single homologous HTH domain, according to a single architecture organization, to thereafter diverge and produce the current TFR superfamily.
 - iv- All in all, there seem to be several factual elements supporting TFR as a genuinely homologous superfamily, which would endorse at least a basic set of common functional mechanisms (albeit with variations built on top, along the diverging evolution).

Again, these comments are just for the sake of this discussion. We don’t believe it is worth including any of these analyses in the paper, needing further elaboration into a separate, evolutionary-oriented piece of work.

- I agree with the authors that an effector/ligand triggering closed to open transition is in opposition with most of the available data. Furthermore, in agreement with the allosteric mechanism presented for FasR, the flexibility of the apo form for many different TetR-family proteins is well documented. By experience, I know also the difficulty to produce crystals of unliganded (apo) TetR-family proteins. A general concept proposes that the apo form is a multi-state form, due to the inherent flexibility in the connection between ligand-binding and DNA-binding domains, whereas the binding with DNA or with effector/ligand traps the protein in a closed or open state, respectively (described for example in Yu et al, 2010, JMB, but also in several other papers). Accordingly, and theoretically, an apo form could be solved in various states, from closed to open state. "Theoretically", because crystallization involves stabilization and reduced flexibility, which is difficult for a protein expected unstable and flexible. Therefore, several crystal structures of TetR-family regulators declared to be in apo form can indeed contain extra electron densities in the ligand-binding cavity, suggesting the presence of unidentified ligands, and consequently these crystal structures are mostly in open dimeric state. Likewise, crystal packings can explain some other apo forms. Nowadays, there are more than 250 structures of a TetR-family protein, with many structures declared in apo form. Did the authors performed an exhaustive survey of the data? From memory, I think PDB id 4JKZ (M. smegmatis Ms6564 in apo form) is in a closed state. Lines 336 to 345 on page 13 should be carefully controlled.

We agree in that such a large protein family represents a challenge in terms of exhaustive analyses.

Although we did perform a survey as comprehensively as we were able to, we cannot of course exclude the possibility of having missed individual examples. We have thus rephrased the referred section on page 13, taking into account the reviewer’s suggestion.

Following up on your specific example, the *M. smegmatis* master regulator Ms6564 was indeed solved in a ligand-free (apo) form as well as bound to DNA (pdb IDs 4JKZ and 4JL3, respectively; doi:10.1074/jbc.M113.468694). However, in this case it is a good example of how crystal packing contacts

can trap a closed-like state of TFRs (we have thus included it in the text, adding it to the ones we gave as examples of this phenomenon, 2FX0, 1T33 and 3VOX). Apo Ms6564 crystallized in a centred orthorhombic space group, diffracting to quite high resolution (1.8Å), consistent with a pretty tight crystalline lattice. Its HTH domain (one monomer in the ASU) is observed burying *ca.* 900Å² due to crystal contacts with neighbouring molecules from other dimers. This is a rather large interface area when it comes to crystal contacts, which gives in this case almost no extra space for the HTH domain to move. Furthermore, although it is a “closed” conformation with regards to the relative position of the HTH and the effector-binding domains, the comparison to 4JL3 reveals a substantial shift away from a true DNA-binding-competent conformation (4.5Å up to ~8Å translational shifts in the main chain trace, including the position of helix α3, combined with a rigid body rotation).

- Five years ago, I analyzed ligand-binding cavities of the TetR-family structures with the aim to compare them (unpublished data). Immediately I was confronted with the difficulty of such an analysis due to the great diversity in cavities but also in the ligand binding sites. Indeed, all the ligands do not bind in the cavity. Thus, I think we have to be careful in claiming a common allosteric mechanism for the TetR family of regulators. I am not saying it is not possible. I am just asking for more experimental evidences. The heterogeneity of TetR-family proteins should prompt us to prudence for a common allosteric mechanism (Colclough et al. 2019, BMC Genomics).

We understood the point being made, and we believe that with the modifications described in response to the previous two comments, now the paper is more cautious and does not make general conclusions for the entire TFR family. Alternative scenarios are now highlighted, all of which will need further investigation with other members of the TetR family of regulators to produce unambiguous interpretations at a more general level.

REVIEWERS' COMMENTS fourth round:

Reviewer #2 (Remarks to the Author):

I am satisfied with the corrected version of the manuscript and would like to recommend the manuscript for publication. Overall, the authors have done a great job addressing the concerns of reviewers. Finally, I thank the authors for the interesting discussion about the homology of the entire TetR family during the reviewing process.